# What Is the Evidence That Dietary Macronutrient Composition Influences Exercise Performance? A Narrative Review

**DOI:** 10.3390/nu14040862

**Published:** 2022-02-18

**Authors:** Timothy David Noakes

**Affiliations:** Department of Applied Design, Cape Peninsula University of Technology, Cape Town 8000, South Africa; noakes@iafrica.com

**Keywords:** muscle glycogen, liver glycogen, hypoglycaemia, fatigue, endurance, diet, carbohydrates, fats

## Abstract

The introduction of the needle muscle biopsy technique in the 1960s allowed muscle tissue to be sampled from exercising humans for the first time. The finding that muscle glycogen content reached low levels at exhaustion suggested that the metabolic cause of fatigue during prolonged exercise had been discovered. A special pre-exercise diet that maximized pre-exercise muscle glycogen storage also increased time to fatigue during prolonged exercise. The logical conclusion was that the athlete’s pre-exercise muscle glycogen content is the single most important acutely modifiable determinant of endurance capacity. Muscle biochemists proposed that skeletal muscle has an obligatory dependence on high rates of muscle glycogen/carbohydrate oxidation, especially during high intensity or prolonged exercise. Without this obligatory carbohydrate oxidation from muscle glycogen, optimum muscle metabolism cannot be sustained; fatigue develops and exercise performance is impaired. As plausible as this explanation may appear, it has never been proven. Here, I propose an alternate explanation. All the original studies overlooked one crucial finding, specifically that not only were muscle glycogen concentrations low at exhaustion in all trials, but hypoglycemia was also always present. Here, I provide the historical and modern evidence showing that the blood glucose concentration—reflecting the liver glycogen rather than the muscle glycogen content—is the homeostatically-regulated (protected) variable that drives the metabolic response to prolonged exercise. If this is so, nutritional interventions that enhance exercise performance, especially during prolonged exercise, will be those that assist the body in its efforts to maintain the blood glucose concentration within the normal range.

## 1. Introduction

### Scientists May Use Different Biological Models to Interpret the Same Experimental Findings

All truths are model dependent. Scientists can and do use different models to interpret the common information that is available to all. As a result, when interpreting the same information, different credible, ethical, objective and trustworthy scientists can come to completely opposite conclusions. Failure to appreciate this phenomenon is the most likely reason why many scientists struggle to understand how colleagues can draw opposing conclusions from their own when all interpret exactly the same experimental data. If we as research scientists are ever to appreciate our own individual intellectual biases, we must each first understand the biological models that we subconsciously use to interpret the scientific information available to all.

The debate about what is the optimum diet for human exercise performance provides a classic example of how our beliefs are fashioned by the models we subconsciously use to interpret the vast body of published scientific literature on this topic. I argue that this example provides a classic confirmation of the statement that ‘what we believe, determines what we believe’.

In this narrative review, I wish to address what is clearly established and what are the model-dependent presumptions of the potential effects that different diets, either high or low in carbohydrate content, have on human exercise performance.

I begin by reviewing the key findings of a series of benchmark, indeed iconic studies that have had an inordinately profound influence in developing the most popular model of human exercise physiology as it relates to diet and exercise performance. That model is so influential that it continues to dominate the interpretation of all subsequent studies, even to the present day.

## 2. The Original and Iconic Studies of the Effects of Diets Differing in Their Macronutrient Contents on Exercise Performance

### 2.1. Studies Prior to the Development of the Needle Muscle Biopsy Technique for the Measurement of Muscle Glycogen Concentrations

In 1920, Krogh [1] perfected the technique for measuring, with extreme precision, the oxygen and carbon dioxide concentrations in air expired by exercising subjects. This allowed he and Lindhard to establish that both carbohydrates and fats are oxidized for energy during exercise [2,3]. Subsequent research showed that the energy contribution from protein metabolism during exercise is small [3,4]. To produce higher rates of fat oxidation during exercise, Krogh and Lindhard had some of their subjects eat foods rich in fat and low in carbohydrate for different periods prior to exercise testing. They reported that in some subjects ‘on (high) fat diets the fatigue (during exercise) became considerable and sometimes excessive. For several hours after the work on the ergometer these subjects were generally very tired when on a fat diet and much less tired or not tired at all when on carbohydrates’ [2]. Surprisingly, other subjects ‘failed to observe any appreciable difference between work on different diets. They never became very tired, either during or after the work’.

Their hypothetical explanation exposed the intellectual model that Krogh and Lindhard subconsciously used to interpret their experimental findings: ‘The variations in fatigue may for instance be correlated with individual differences in the liability to acidosis. The clearing up of this point will require a special investigation in which the hydrogen ion concentration of the blood will have to be determined during and after the work’.

Krogh and Lindhard clearly favoured an exercise model in which the high-fat diet causes skeletal muscular acidosis, which is then the direct cause of their high levels of fatigue both during and after exercise. Their model was probably based on the dominant model of fatigue then popular; one developed by English Nobel Laureate A.V. Hill in the early 1920s to explain how fatigue develops during exercise of high intensity [5]—what has since been termed Hill’s Cardiovascular-Anaerobic model [6]. Hill’s model continues to enjoy a committed following nearly a century later [7,8]. It proposes that a skeletal muscle anaerobosis developing during high-intensity exercise causes a progressive rise in muscle and blood lactate concentrations, which ultimately limit exercise performance [9,10,11].

The next advance in the study of metabolism during especially prolonged exercise came with the development of the laboratory treadmill.

In 1934, Edwards, Margaria and Dill [12] reported the results of their ‘ablest subject’ runner Y ‘aged 19, a skilled runner who had finished near the front in marathons’ when he ran for up to 6 h on a laboratory treadmill at speeds of between 9.3 and 11.3 km/h on 3 separate occasions.

Figure 1 shows that these studies established that Y’s blood ‘sugar’ (glucose) fell progressively during exercise, reaching a value consistent with hypoglycemia (<70 mg/100 mL; 3.9 mmol/L) after 6 h. His respiratory quotient (RQ) also fell progressively, indicating a gradual shift from carbohydrate to fat metabolism. After 6 h of exercise, fat metabolism accounted for approximately 80% of Y’s energy use. Y used approximately 375 g of carbohydrate during the exercise bout.

The authors concluded that ‘as carbohydrate reserve diminishes the proportion of energy derived from fat may increase from 8 per cent to 77 per cent. There is no evidence that carbohydrate is more essential in work than in rest’ [12].

Perhaps predictably, the first comprehensive studies of the effects of diets, differing in their macronutrient compositions on exercise performance, were conducted in Krogh and Lindhard’s laboratory in Copenhagen by those whom Krogh and Lindhard had directly influenced—Christensen, Asmussen and Nielsen [13,14]. They would add two classic discoveries (Figure 2 and Figure 3).

First, they confirmed Krogh and Lindhard’s observation that exercise performance was greatly impaired in subjects who had eaten a low-carbohydrate diet prior to the exercise bout. It is important to note here that this study was an acute (less than 7 days) dietary intervention. This becomes an important consideration subsequently.

Figure 2 shows that when eating the low-carbohydrate diet, subjects could exercise for less than 90 min at a work rate they could sustain for up to 240 min when eating the high-carbohydrate diet.

Second, blood glucose concentrations fell progressively during exercise in the low-carbohydrate group, reaching hypoglycaemic levels already within the first 30 min of exercise (Figure 2).

Clearly, the authors understood the importance of this finding because in their next series of experiments, they had subjects ingest 200 g of glucose at the point of fatigue. We can conclude that their goal was to test whether reversal of hypoglycemia would allow the subjects to continue exercising.

Figure 3 shows that two subjects who ingested 200 g of glucose at the point of fatigue were able to continue exercising for another hour. In both, glucose ingestion rapidly normalized their low and falling blood glucose concentrations. Oxygen consumption (VO_2_) and RQ increased in one subject (bottom panel), indicating an increased work output.

The most reasonable explanation of the findings of these two early studies should have been that prolonged exercise causes the blood glucose concentration to fall, leading to the termination of exercise when its concentration falls below approximately 70 mg% (70 mg/100 mL; 70 mg/dL; 70 mM; 3.9 mmol/L), causing symptomatic hypoglycemia. Ingestion of glucose reverses the hypoglycemia, allowing exercise to continue for an appreciable additional period. The authors would therefore have concluded that hypoglycemia is an important cause of fatigue during prolonged exercise but that this can be reversed acutely by glucose ingestion.

Hence, the conclusion would have been that these original studies established that if exercise is sufficiently prolonged to induce hypoglycemia, carbohydrate ingestion can reverse that hypoglycemia-induced fatigue, allowing exercise to continue.

### 2.2. Studies That Followed the Adoption of the Needle Muscle Biopsy Technique for the Measurement of Muscle Glycogen Concentrations

Not unexpectedly, Scandinavian researchers produced the next major advance in this field by adopting the Duchenne needle muscle biopsy technique, which allowed the measurement of glycogen content in human skeletal muscle for the first time [15]. This innovation created the single greatest intellectual revolution in the history of the exercise sciences.

However, it was not all beneficial because it influenced many, myself very much included [16], subconsciously to adopt a reductionist approach to human exercise performance. We began to believe that human exercise performance could be explained by a single variable, in this case the athlete’s pre-exercise muscle glycogen content. All other factors suddenly became largely irrelevant, thus much less interesting to consider and to study.

A clear analogy is the reductionistic model A.V. Hill used to explain fatigue during high-intensity exercise. Even though Hill was a Nobel Laureate, he developed his (reductionist) model within the constraints imposed by the limitations of what he could realistically measure in exercising humans in the 1920s. Essentially his only options, other than heart rate, also measured with difficulty, were blood lactate concentrations and rates of oxygen consumption. Within these constraints, Hill’s mind conceived a model of exercise performance based solely on these two variables [6,9,10,11].

Similarly, the ability finally to measure muscle glycogen content encouraged the reductionist belief that exercise performance, especially during prolonged exercise, could henceforth be explained by the single biochemical that this novel technique allowed to be measured for the first time in exercising humans.

For their iconic study, Bergstrom, Hermansen, Hultman and Saltin [17] studied a group of nine subjects during and after an exercise bout, which followed three different 3000 kcal diets comprising high- (82% carbohydrate (CHO), 10% fat, 8% protein); mixed- (52% CHO, 27% fat, 21% protein) or low-carbohydrate (5% CHO, 54% fat, 41% protein) content. The important results are shown in Figure 4.

Figure 4 shows that starting muscle glycogen content was a function of the pre-exercise diet and was highest in subjects when they ate the high-carbohydrate diet and least following the low-carbohydrate diet. The exercising RQ was also higher in the high- and mixed- carbohydrate diets. In all three trials, blood glucose levels fell progressively, reaching hypoglycaemic concentrations (<70 mg/dL) after 30, 60 and 90 min in the different trials. Hypoglycemia occurred much earlier and to a far greater extent in the group eating the low-carbohydrate diet.

The authors also provided a figure, redrawn here as Figure 5, showing a linear associational relationship between starting muscle glycogen concentration and the duration for which exercise could be sustained at an intensity of 75% VO2max after the three different diets.

They concluded the following: ‘The good correlation between initial glycogen concentration and work time (shown here in Figure 5—current author’s addition) demonstrates that the individual’s ability to sustain prolonged exercise is highly dependent on the glycogen content of the muscles which, in turn, is dependent on the type of diet before exercise’ [17].

Subsequently, they neatly framed the hypothesis that has been handed down ever since as verified fact to subsequent generations of exercise scientists and sports nutritionists: ‘The present results reveal that the glycogen content of working muscle falls steadily during exercise. In addition they show that the store of muscle glycogen can be emptied, and that the ability to continue exercise with the same load is limited when the glycogen concentration approaches zero’ [18]. Interestingly, they also noted that intravenous glucose infusions have little effect on the rate of muscle glycogen utilization: ‘…sugar (sic) supplied by supplied by the bloodstream is able, only to a minor extent, to replace muscle glycogen as a source of energy during this type of muscular work’ so that ‘our data suggest that (during glucose infusion) the glycogen utilization continues on an inappreciably reduced scale, despite the availability of glucose’.

I will argue that the discovery that muscle glycogen use is not inhibited by glucose infusion (or ingestion) during prolonged exercise is a crucially important finding, the true significance of which has largely been ignored. However, its relevance to a proper understanding of the metabolic response to exercise and especially the role of different diets on that metabolic response can, I suggest, no longer be ignored.

Interestingly, the authors ignored the low blood glucose concentrations in these athletes at the end of exercise (Figure 4) preferring to focus their attention on the muscle glycogen changes as the exclusive cause of fatigue, which is to be expected since the innovation they had pioneered was the ability to measure muscle glycogen, not blood glucose concentrations. Thus, being human, they were naturally biased towards conclusions that would advance the importance of their novel innovation. Their bias in turn made it more likely that, from the moment they applied the muscle biopsy technique, these authors would promote a model of human performance in which muscle glycogen would be the key determinant of performance during prolonged exercise [15,17,18,19,20,21,22,23,24].

The authors also noted that exercise terminated at different final muscle glycogen concentrations, which were highest after the high-carbohydrate diet. This might suggest that there is not one single ‘limiting’ muscle glycogen concentration at which fatigue always develops.

Bergstrom et al.’s major conclusion was that ‘the glycogen content of the working muscle is a determinant for the capacity to perform long-term heavy exercise. Moreover, it has been shown that the glycogen content and, consequently, the long-term work capacity can be appreciably varied by instituting different diets after glycogen depletion’ [17].

Yet the study did not establish that the glycogen content of the working muscles is a ‘determinant’—that is a direct cause—of the duration for which ‘long-term heavy exercise’ can be performed. The study simply established an association between the pre-exercise muscle glycogen content and exercise duration. It did not exclude the possibility that diet influenced not only muscle glycogen concentrations but also other variables, which might be the true, albeit the unrecognized factor(s) determining fatigue.

One such unrecognized factor would be the very low blood glucose concentrations at the end of exercise. Importantly at the point of exercise terminaton, these concentrations were lowest after the low-carbohydrate diet and highest after the high-carbohydrate diet (Figure 4). Indeed blood glucose concentrations were already in the hypoglycaemic range within 30 min of starting exercise after the low-carbohydrate diet (Figure 4).

Proving that a low muscle glycogen content ‘determines’ the exercise duration by fixing the moment of exercise termination requires evidence that increasing muscle glycogen content alone at or near the point of exhaustion can reverse fatigue, allowing exercise to continue. However, presently there is no known intervention that can acutely increase muscle glycogen concentrations. Since this is not possible, definitive proof that very low muscle glycogen concentrations are the single, direct cause of exercise termination, is unobtainable.

However, the conclusion that a low muscle glycogen content is the direct cause of exercise termination during prolonged exercise has been embraced by the scientific community as proven fact ever since the publication of this study with its data depicted in Figure 5. The corollary, relevant to this review, is the universally accepted belief that any dietary intervention that increases pre-exercise muscle glycogen content must increase performance during prolonged exercise; the opposite applies for interventions that reduce these concentrations.

Despite these considerations, another iconic researcher of the era, Professor Per-Olaf Astrand [23], affirmed these conclusions: ‘With high workloads, demanding 75% or more of max VO_2_, it seems that the glycogen content in the exercising muscles will be an important determinant of maximal work time…We thus conclude that different diets can apparently influence markedly the glycogen stores in the muscles. The ability to perform *heavy, prolonged* exercise is correspondingly affected and the higher the glycogen content, the better the performance…When the glycogen stores are exhausted, it seems impossible to work at the same high rate as with glycogen available’.

Interestingly, Astrand did consider the findings of Christensen and Hansen [13,14] concluding that: ‘Thus, it appears that one limiting factor in exhaustive work of this kind is the fall in blood sugar, and that the nerve cells suffer more than the muscle cells’.

However, history has tended to ignore this statement in favour of the belief that the size of the pre-exercise muscle and not liver glycogen stores is the ultimate determinant of exercise performance.

The next iconic study of this era was a clinical trial comparing the effects of high- carbohydrate and mixed diets in athletes running the same 30 km race [19] on two occasions, three weeks apart. Figure 6 shows that after the high-carbohydrate diet, athletes began with muscle glycogen concentrations that were twice as high as when they ate the mixed diet; they also completed the 30 km race approximately 8 min faster following the higher carbohydrate diet. Muscle glycogen concentrations following the mixed diet reached very low levels at the end of the race.

There was also a linear associational relationship between the starting muscle glycogen concentrations and the reduction in performance following the mixed diet (data not shown).

Figure 7 shows the individual performance of the athletes at seven check points, each 4 km apart, during the race and at the finish. It records the difference in running times between the two trials at these eight check points and relates this to the runners’ predicted muscle glycogen concentrations at those check points.

Figure 7 shows that the six identified athletes who started the 30 km race after the mixed diet with the lowest muscle glycogen concentrations—all below 15 g/kg—slowed the most in the second half of the race. In contrast, the other four athletes, with starting muscle glycogen concentrations greater than 19.0 g/kg after the mixed diet, slowed the least.

Thus, the conclusion from Figure 7 was that the athletes who started the race with the lowest muscle glycogen concentrations showed the most marked slowing in the second half of the race. Indeed, the four athletes with the lowest pre-exercise muscle glycogen concentrations following the mixed diet (BSt, RG, JS and BSj) slowed by more than 10–15 min compared to when they were ‘carbohydrate-loaded’.

Importantly, the authors did not report any measurements of blood glucose concentrations in these athletes. However, from the evidence that will be presented from other studies, it can be predicted with absolute certainty that those athletes, who began the race after the mixed diet with the lowest muscle glycogen concentrations, would also have shown the earliest and largest reductions in their blood glucose concentrations during the 30 km race. However, because the authors designed the experiment to test their evolving model that muscle glycogen concentrations alone determine endurance performance, they ignored this probability. As a result, this concern has not been communicated to subsequent generations of exercise scientists. So, it has gone missing from the debate pretty much ever since.

Predictably, the study evoked a tsunami of interest since the findings appeared to be so obvious. By eating a high-carbohydrate diet before exercise, some athletes could expect to run as much as 15 min faster in a 30 km race compared to when they ate the lower-carbohydrate, mixed diet that had, up until that point in athletic history, been the more usual choice of elite athletes. This promise of a simple method to achieve such a boost in performance rapidly captured the attention of all the world’s leading endurance athletes, as well as the world’s most influential exercise scientists, sports doctors and dietitians.

In order to maximize their pre-race muscle glycogen concentrations in this study, the subjects had undergone what became known as the ‘Saltin diet’, named after the senior author.

In the ‘Saltin diet’, athletes were encouraged to exercise for 2 h, 7 days before their upcoming race. For the following three days, they continued to exercise but their diet contained no or little carbohydrate. During this period, athletes continued to exercise as best they could, attempting to completely deplete their muscle (and liver) glycogen stores. Then for the final three days before the race, subjects rested and were encouraged to eat at least ‘2500 kcal/day’ of carbohydrates. This would require an intake of 625 g of carbohydrate.

In fact, the ‘Saltin’ diet was first described by Ahlborg et al. [22], who showed that this dietary/exercise regime could increase muscle glycogen concentrations ~50% more than did eating the same high-carbohydrate diet for a single day.

The popular conclusion from these two iconic studies was that the pre-exercise muscle glycogen content is the pre-eminent determinant of human exercise performance [21,23]; that pre-exercise concentrations can be maximized by a specific diet/exercise regime [22,23]; that this procedure substantially improves human exercise performance because the cause of fatigue and of ‘slowing down’ (Figure 7) in more prolonged exercise is the near-total depletion of muscle glycogen stores.

However, it is important to note that the 30 km running experiment [19] was not conducted as a randomized controlled trial, nor was it placebo-controlled. Nor were conditions identical for both 30 km races. One of these races was run in the company of 1500 other competitors; in the second “race”, there were just the 10 subjects. Thus, this should be remembered as a hypothesis-generating study, not as definitive proof for any particular hypothesis.

Whilst the data in Figure 7 are remarkably compelling—suggesting the severely detrimental impact on performance of low muscle glycogen concentrations—I argue that they might equally have been explained by differences in blood glucose concentrations during the race, measurements that the authors did not consider because of their intellectual biases, which produced the specific hypothetical fatigue model they were testing. Yet Bergstrom and Hultman [24] did subsequently acknowledge that the carbohydrate-loading diet ‘also increases the liver glycogen store’.

It is important to remember that a difference in starting muscle glycogen concentrations is not the sole change produced by the high-carbohydrate diet. There will also be differences in liver glycogen concentrations [25] and likely also in brain glycogen concentrations [26].

## 3. Modern Studies to Determine the Mechanisms by Which Carbohydrate Ingestion Either for the Entire Duration of the Exercise Bout or First Taken at the Point of Exhaustion Can Delay or Reverse Fatigue

Three additional questions were addressed by two other relevant studies of the era.

Christensen and Hansen had clearly shown that when given at the point of exhaustion during prolonged exercise, glucose ingestion could reverse fatigue (Figure 3). This was confirmed by the study of Coggan and Coyle [27], who, using a more sophisticated research design (Figure 8) than the Scandinavians, sought to explain the mechanisms for the effect identified by Christensen and Hansen.

Coggan and Coyle [27] exercised 7 cyclists for 3 h at 70% VO2max on three separate occasions. The cyclists then rested for 20 min before commencing a further exercise bout to exhaustion. Before or during this second exercise bout, subjects received either a placebo, or a carbohydrate drink, or an intravenous glucose infusion, the latter returning the blood glucose concentration to the normal range.

Figure 8 shows that the initial bout of exercise produced marked hypoglycemia in all three trials; indeed, a majority of the cyclists reported that they developed hypoglycaemic symptoms. However, the authors possibly underestimated the relevance of these symptoms, perhaps also hinting at their own intellectual bias based on the model of exercise performance that they prefer: ‘Most of the subjects experienced mild central nervous system symptoms during the final 30–60 min of exercise… (but) These symptoms did not appear severe enough, however, to be the sole cause of the subjects’ inability to continue exercise’ [27].

In an earlier, exploratory study, Coyle et al. [28] showed that they considered that only symptoms of severe hypoglycemia would likely interfere with exercise performance. In a study of 7 subjects, four finished with average blood glucose concentrations of 2.5 mmol/L: ‘Only two subjects developed symptoms suggestive of hypoglycemia, which included light-headedness, weakness, disorientation, and nausea’. Blood glucose concentrations at the time of exercise termination were, however, profoundly hypoglycaemic—2.5 and 2.0 mmol/L. The possibility exists, as I will argue, that a falling blood glucose concentration is a sufficient signal to cause a reduction in exercise intensity, well before significant hypoglycemia is measured in the blood.

In their later study [27], time to fatigue in the second exercise bout was clearly determined by the reversal of hypoglycemia and was shortest in the placebo trial; somewhat longer following glucose ingestion; but extended by 45 min with intravenous glucose infusion. Importantly, glucose infusion reversed the hypoglycemia and increased the RER but did not influence muscle glycogen use during the second exercise bout. In fact, muscle glycogen concentrations did not decrease during the second exercise bout in any of the conditions (data not shown).

This confirms that subjects did not terminate exercise at the end of either the first or second exercise bout because they had reached a limiting, low muscle glycogen concentration. Instead, the development of an uncorrected hypoglycemia is the more probable explanation for the termination of exercise in the placebo- and glucose-ingestion groups. Since cyclists in the glucose infusion group were not hypoglycaemic when they terminated the second exercise bout, since their RER was not low (0.87; Figure 8) and their rate of carbohydrate infusion was high (1.1 g/min), it is not possible to provide any purely metabolic explanation for their fatigue based on what was measured in these experiments and on the basis of the hypothetical (muscle glycogen) model used by these authors to explain fatigue.

The authors [27] concluded that: ‘the current results clearly demonstrate that the decline in plasma glucose that often accompanies prolonged exercise is the fasted state can contribute to the development of fatigue. The inability to continue exercise was closely related to the lowering of plasma glucose and subsequent fall in carbohydrate oxidation, suggesting that fatigue was due to an inadequate supply of carbohydrate for the working muscles. This decline in carbohydrate oxidation could be reversed and exercise continue for ~45 minutes’ when glucose was infused intravenously.

It is again important to note that whilst the authors acknowledge that hypoglycemia was clearly a factor limiting exercise performance, their explanation of how glucose infusion reverses that limitation is based on the unproven but universally-accepted muscle glycogen/carbohydrate depletion model which holds that muscle glycogen must provide an obligatory carbohydrate contribution to skeletal muscle energy metabolism without which exercise cannot be sustained.

Thus, according to these authors, the provision of carbohydrate either by infusion or ingestion provides an alternate source of carbohydrate to sustain this obligatory need for carbohydrate oxidation. Since their model is ‘brainless’ [11], it allows no room for an alternate explanation, specifically that hypoglycemia poses the gravest threat to the health of the human brain [29], so complex homeostatic mechanisms must exist to prevent the development of critical hypoglycemia, especially during exercise [30].

The most effective mechanism during exercise would be simply to reduce or limit and ultimately terminate any central motor command to the exercising limb muscles [11]. This would instantly reduce muscle glucose oxidation, allowing the blood glucose concentration to rise within seconds to minutes, as occurs normally in those who terminate exercise with normal blood glucose concentrations [31].

An analogy is the control exerted by the brain and central nervous system in terminating exercise, particularly at extreme altitudes, to prevent the development of a catastrophic brain hypoxia [6,9,11]—the so-called ‘lactate paradox of high altitude’ [32].

In his extensive, remarkably comprehensive and brilliant 1994 review of the cellular mechanisms of fatigue, R.H. Fitts [33] also reviewed the way in which the original iconic Scandinavian studies have been interpreted but perhaps with too little skepticism. For, his balanced conclusions have largely been ignored by the scientific community, especially by those who promote a model in which muscle glycogen has an obligatory role in skeletal muscle metabolism—particularly during prolonged exercise. Fitts’s cautious conclusions need to be considered.

‘It became clear from these early studies that carbohydrate supplementation delayed the onset of fatigue during prolonged endurance exercise. However, the mechanism of the protective effect was unknown. Carbohydrate supplementation could improve performance by preventing low blood sugar and the development of neuroglycopenia or, alternatively, the important factor could be the continued high supply of a carbohydrate fuel [34]. The latter would be particularly important in the final stages of prolonged exercise when muscle glycogen stores were low or depleted. Over the past 25 years, this question has been extensively studied, and the preponderance of evidence supports the notion that carbohydrate ingestion delays fatigue by maintaining a high carbohydrate fuel source in the form of blood glucose [34,35]. This conclusion necessitates the hypothesis that carbohydrate oxidation is essential to the maintenance of prolonged exercise at moderate to high intensity [65–90% of one’s maximal oxygen uptake (VO2max)]. To date, a cellular explanation for an obligatory oxidation of carbohydrates has not been established (current author’s added emphasis). The possibilities include the following: (1) high muscle oxidation rates cannot be maintained without a carbohydrate fuel source, (2) carbohydrates supply critical metabolic intermediates such that carbohydrate depletion reduces the oxidation rate of available fats and proteins, or (3) carbohydrate depletion is correlated with (and perhaps causative of) changes in other cellular events that in turn elicit fatigue. Little direct support exists for any of these possibilities ((current author’s added emphasis). The strongest evidence linking any of these possibilities to muscle fatigue is a consistent observation that fatigue during endurance exercise is associated with glycogen depletion [17,18,21,36]’.

To which one might add, glycogen depletion in either or both muscle and liver (current author’s added emphasis) and perhaps also in brain glycogen content [26,37,38].

Later in the same article, Fitts again emphasized that: ‘The exact mechanisms of the protective effect of carbohydrate oxidation in the prevention of fatigue are unknown… The energy derived from the oxidation of fats per liter of oxygen consumed is less than that obtained from carbohydrate oxidation. However, this difference is small and would not explain the complete exhaustion frequently associated with prolonged exercise’.

Importantly, Fitts also provides two studies in which muscle fatigue was dissociated from both muscle and liver glycogen depletion [39,40], indicating that other mechanisms [33,41] must be involved.

Fitts’s review establishes that, at the time he wrote the article in 1994, there was no evidence proving an obligatory need for carbohydrate or, more specifically muscle glycogen oxidation necessary to sustain performance during prolonged exercise. This is a critical conclusion that continues to be conveniently ignored. As I will show, since publication of this review, two carefully conducted studies of the highest quality have clearly disproven this theoretical explanation [42,43] although one [43] does not interpret the findings in the way I suggest.

In writing his review, Fitts chose specifically to focus on possible peripheral mechanisms of fatigue and to avoid discussion of the relative importance of central (brain) versus peripheral (muscle) mechanisms in the fatigue process.

However, by excluding any potential contributions of central (brain) mechanisms, one is left with a ‘brainless’ model of exercise performance [11] which ignores the importance of the homeostatic regulation of exercise by the central nervous system [44,45], and the likely influence of factors like hypoglycemia on this homeostatic control mechanism.

For example, the hypothesis that there is this obligatory need for carbohydrate oxidation during exercise is based largely on the original evidence that carbohydrate ingestion delays or reversed fatigue during prolonged exercise [13,14] and that fatigue develops prematurely in those who begin exercise in a carbohydrate-depleted state [17,18,19].

However, as should be clear from Figure 2, Figure 3, Figure 4 and Figure 8, hypoglycemia was always present whenever fatigue developed in each of these studies. Thus, an equally plausible explanation is that the ‘obligatory’ carbohydrate is not required to maintain the optimum functioning of the exercising muscles (through an obligatory contribution to their carbohydrate metabolism) but an obligatory need to maintain blood glucose concentrations to sustain normal cerebral functioning. Higher starting liver glycogen concentration as a result of ‘carbohydrate loading’ would partially achieve this effect.

Figure 9 provides an example of the complex central (brain) mechanisms involved in regulation of the blood glucose concentration [46]. The model shows that multiple peripheral metabolic signals convey information to the hypothalamus and brain stem, which, via the autonomic nervous system (ANS), modulate pancreatic insulin/glucagon secretion, hepatic glucose production and skeletal muscle glucose uptake (current authors’ added emphasis).

Importantly, there is one crucial element missing in the model depicted in Figure 9. In their model, Roh et al. [46] have failed to consider the complicating influence of prolonged exercise on homeostatic regulation of the blood glucose concentration. So missing from their diagram is a necessary link between these hypothalamic and brain stem centers and the motor cortex specifically to prevent excessive skeletal muscle recruitment during prolonged exercise when liver glycogen depletion develops causing hypoglycemia. Without this added control, all forms of prolonged exercise would ultimately cause hypoglycaemic brain damage.

So, whilst the authors acknowledged that for their model to work under resting conditions, skeletal muscle glucose uptake needs to be regulated centrally by the autonomic nervous system, they ignore the single greatest driver of skeletal muscle glucose uptake especially during exercise, which is the extent of skeletal muscle motor unit recruitment in the exercising limbs, directed by the motor cortex in the cerebral frontal lobe.

For this model to also work during prolonged fatiguing exercise, there must be a mechanism by which these control centers in the hypothalamus, brain stem and autonomic nervous system can regulate, on a moment-to-moment basis, the levels to which the motor cortex in the brain’s frontal lobe is recruiting motor units in the exercising skeletal muscles. Thus, there must also be a link between the brain centers depicted in Figure 9 and the motor control centers in the cerebral cortex.

Recognizing the existence of this feedback–feedforward crosstalk mechanism between these centers in the hypothalamus and brainstem, and the motor control centers in the cerebral cortex, allows the following predictions.

Normoglycemia will allow the maintenance of central motor drive to the exercising limb muscles, so that the exercise can continue at the required intensity. However, the development of a progressive hypoglycemia will cause a gradual reduction in central motor drive to the exercising muscles, producing an irreversible reduction in the exercise intensity that the athlete can willingly sustain [44]. This reduction in exercise intensity with the onset of a progressive hypoglycemia is clearly shown in Figure 10 and in a number of studies to be reviewed subsequently (see Figures 20–24 and 26).

However, if only a ‘brainless’ model of exercise performance is considered [11], the absolute essentiality of this feedforward control of skeletal muscle motor unit recruitment to prevent hypoglycaemic brain damage [29] during prolonged exercise will not be recognized.

Instead, the “brainless” model permits only one interpretation of the currently published evidence, specifically that there is an obligatory requirement for carbohydrate/muscle glycogen oxidation to sustain prolonged exercise. So, it is that what we believe—in terms of the models we use to explain the phenomena we observe—determines what we believe.

What remained to be resolved in the late-1980s was whether glucose ingestion during prolonged exercise but prior to the onset of exhaustion could enhance performance and, if so, what were the possible mechanisms involved. Although there were numerous unsuccessful attempts to answer this question [28,47,48], a final definitive answer was again provided by Coyle et al. [49] in another iconic study published in 1986. The authors exercised 7 highly-trained cyclists at 71%VO2max for as long as they could last. Subjects began exercise after they had fasted for 16 h prior to exercise; there is no record that they were given any other specific dietary advice.

**Figure 10 nutrients-14-00862-f010:**
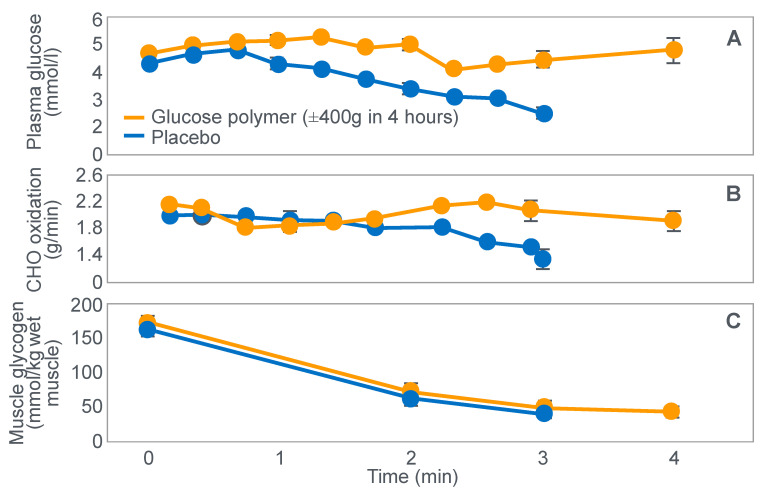
Changes in plasma glucose concentrations (panel **A**), rates of carbohydrate (CHO) oxidation (panel **B**) and muscle glycogen concentrations (panel **C**) during prolonged exercise when subjects ingested either placebo or 100 g of carbohydrate every hour during exercise. Subjects began the trial after a 16 h fasting period. Adapted with permission from ref. [49]. Copy-right 1986 American Physiological Society.

During one exercise bout, subjects ingested 100 g of a glucose polymer drink every hour during exercise; in the other, they ingested a sweetened placebo. Thus, properly described, the trial examined the effects of carbohydrate ingestion during exercise by subjects who began exercise with normal muscle glycogen concentrations but with acute, fasting-induced liver glycogen depletion. This detail that has been generally overlooked in the decades since the study was reported.

Figure 10 shows the important results of the study.

Figure 10 shows that when they exercised without glucose ingestion, blood glucose concentrations fell to ~2.5 mmol/L within the first 3 h of exercise, causing profound hypoglycemia, whereas glucose ingestion during exercise maintained blood glucose concentrations nearer to 5 mmol/L. Importantly, glucose ingestion did not alter muscle glycogen use during exercise in those who began exercise with normal or elevated muscle glycogen content, as has been shown repeatedly [50,51,52,53,54,55,56,57,58]. Only when muscle glycogen levels reach low levels during the final hour of more prolonged exercise does it appear as if carbohydrate ingestion might spare muscle glycogen use [59,60].

The key point is that although glucose ingestion allowed subjects to exercise for an hour longer in this trial, they nevertheless terminated exercise after 4 h with the same muscle glycogen concentrations as they had when they exercised without glucose ingestion but chose to stop exercising after 3 h (Figure 10). Thus, the additional hour of exercise happened without any need for an ‘obligatory’ ATP generation from muscle glycogenolysis.

As the authors stated: ‘highly trained endurance athletes are capable of oxidizing carbohydrate at relatively high rates from sources other than muscle glycogen during the latter stages of prolonged exercise and this postpones fatigue’. So that ‘…when blood glucose concentration was maintained, highly trained endurance athletes were capable of oxidizing carbohydrate sources other than muscle glycogen at high rates during the latter stages of prolonged strenuous exercise’.

The authors careful use of language allowed them to ignore a rather more obvious conclusion, specifically that it was not their low muscle glycogen concentrations that caused subjects of fatigue in the control condition, when they did not ingest any carbohydrate during exercise, but rather their inability to maintain their blood glucose concentration above ~4 mmol/L.

This finding returns us to the critical question raised by Fitts [33]: What is it about carbohydrate metabolism that is essential for sustaining prolonged submaximal exercise performance? Is there indeed such an essential need? Why is it not possible that an increased ATP provision from fat oxidation could substitute for ATP generated by carbohydrate oxidation when muscle glycogen stores are depleted? Must this (brainless) model of exercise performance, regulated exclusively in the periphery, remain unchallenged because of our uncritical devotion to the original Scandinavian studies [17,18,19,20,21,22,23,24], none of which conclusively proved that the metabolism of muscle glycogen generates an obligatory pool of metabolic intermediates and a special pool of ATP, without which prolonged exercise is unsustainable?

An important question, as also identified by Fitts [33], is why the subjects in the study shown in Figure 10 chose to stop exercising exactly at 4 h (following the third muscle biopsy) when they were ingesting carbohydrate which maintained their blood glucose concentrations and rates of carbohydrate oxidation and when their muscle glycogen concentrations were no lower than they had been an hour earlier. One likely explanation is that their ratings of perceived exertion (RPE) were extremely high (18 units) at exercise termination, suggesting that their decision to terminate exercise may have been due to a central (brain) interpretation of what was happening in the peripheral muscles.

It is important once more to stress that this was a study of subjects who began exercise with normal or elevated muscle glycogen concentrations but liver glycogen stores depleted by 16 h of fasting. Thus, the unequivocal demonstration of a measurable ergogenic effect of carbohydrate ingestion on exercise performance seems to require that subjects fast for at least 12–16 h before exercise. This makes it more likely that hypoglycemia will develop and, according to the explanation that I propose, that a progressive fall in the blood glucose concentration must ultimately activate a protective brain-directed reflex to reduce the exercise intensity, in order to avoid the possibility of hypoglycaemic brain damage.

This homeostatic model of exercise regulation by the brain in order to protect against a harmful outcome—hypoglycaemic brain damage—is the opposite of the more usual non-homeostatic ‘limitations’ models [61], one of which hypothesizes that the accumulation or depletion of specific brain chemicals, especially 5-hydroxy tryptamine (5-HT) or perhaps specific cytokines [62], causes exercise termination. However, this explanation is closer to another ‘limitations’ model in which brain glycogen concentrations may also play at least a permissive role in the regulation of prolonged exercise performance [26,37,38].

This homeostatic model proposes that carbohydrate interventions that maintain or normalize the blood glucose concentration [14,27,49] will remove a postulated (but essential) brain-directed homeostatic control, permitting the exercise to continue and perhaps even allowing the exercise intensity to increase [14].

However, if the evidence from these studies is interpreted solely according to a “brainless” model of human exercise performance [11], exactly the same data can only ever be (mis)interpreted as evidence that the sole importance of maintaining normoglycemia is to provide sufficient “obligatory” carbohydrate for oxidation by the exercising muscles.

Instead, it is clear that the brain itself also has an obligatory need for (some) carbohydrate use [63].

## 4. Is the Main Ergogenic Effect of Pre-Exercise Carbohydrate Loading or Carbohydrate Ingestion during Exercise to Alter the Metabolism of the Skeletal Muscles? Or of the Liver?

Another critically important study [64] from Dr. Coyle’s research group found that a constant glucose infusion that maintained the blood glucose concentration at >10 mmol/L also failed to influence muscle glycogen use during 2 h of exercise at 73%VO2max (Figure 11). As already described, the failure of carbohydrate ingestion or infusion to influence muscle glycogen use during exercise is well established. However, no study will ever produce results that are quite as convincing as those shown in Figure 11.

Figure 11 shows that despite a very high rate of glucose infusion (1.6 g/min rising to 2.6 g/min after 120 min of exercise), extremely high plasma glucose and insulin concentrations and high rates of carbohydrate oxidation, the rate of muscle glycogen use was identical in both trials. Weltan et al. [65,66] have also shown that glucose infusions do not reduce muscle glycogen use during prolonged exercise even in subjects who begin exercise with low muscle glycogen concentrations.

The finding of all these studies [49,50,51,52,53,54,55,56,57,58,59,64,65,66] are extraordinarily important since they definitively establish that the rate of muscle glycogen use during exercise is not influenced by carbohydrate ingestion or even infusion at very high rates. Instead, it is now clear that the main effect of carbohydrate ingestion or infusion during exercise is to reduce the rate of hepatic gluconeogenesis, thereby protecting the liver from becoming glycogen depleted, and thus delaying or preventing the development of a progressive hypoglycemia during exercise [50,52,56,60,67]; pre-exercise carbohydrate loading [52,68] aids by reducing the contribution of blood glucose oxidation to overall carbohydrate oxidation, thereby also contributing to reduced hepatic glycogenolysis.

Importantly, pre-exercise ‘carbohydrate loading’ also does not ‘spare’ muscle glycogen use during exercise; rather, it maintains higher rates of muscle glycogenolysis even as concentrations fall (Figure 10 in [52]; Figure 7 in [68]).

The important conclusion must be that glucose ingestion or infusion does not enhance exercise performance [14,27,49,51,57,58,59,69,70,71,72,73] by slowing muscle glycogen use, which in turn means that carbohydrate ingestion during exercise most likely influences exercise performance by a central brain mechanism: either by maintaining central motor drive by protecting against hypoglycemia, or by as yet to be determined reflex central brain mechanism [74,75,76].

The evidence that this effect is more likely due to the prevention of hypoglycemia is suggested by a number of modern studies to be discussed shortly.

## 5. What Is the Model of Exercise Physiology and Fatigue That Developed as a Result of the Original Iconic Scandinavian Studies?

The conceptual model that developed from these iconic Scandinavian studies, most especially the two iconic studies depicted in Figure 4 and Figure 5, is that performance during prolonged exercise is ultimately limited by inadequate carbohydrate oxidation secondary to muscle glycogen depletion. As perhaps the leading applied sports scientist in the world in the 1970 and 1980s, Dr. David Costill, wrote in 1973 [77]: ‘Depletion of the endogenous carbohydrate stores has been shown to be a limiting factor in the ability to perform long-term exercise’. To support his conclusions, he cited the classic Scandinavian studies [13,17,22,78,79].

As already described, Fitts [33] has offered a number of possible explanations, all unproven, of why carbohydrate oxidation serves an obligatory role for sustained exercise performance.

Other highly influential sports scientists have also expressed their opinions of why carbohydrate is an obligatory muscle fuel necessary to sustain prolonged exercise of higher intensity, usually described as exercise at intensities greater than 60–75% VO2max [80,81,82,83,84]. Thus: ‘CHO-based fuels become the predominant energy source for trained muscle when exercise intensities are >60% of peak oxygen uptake’ [80]. This is because ‘endogenous fatty acid stores are substantial but do not provide muscle contractile energy (i.e., adenosine triphosphate) at rates that sustain higher exercise intensity or high force contractions’ [84] and because ‘rates of muscle fat oxidation are inadequate to support the high relative (70–90% VO2max) and absolute work rates sustained by competitive athletes during running or cycling events lasting < 2 h [80,83,85,86,87]. Additionally, ‘high rates of carbohydrate oxidation are required to sustain high-intensity cycling (>80% VO2max)’ [88], so that ‘Clearly at the intensities at which competitive endurance athletes train and race (>70–75% of VO2max), the exercising muscles are dependent on carbohydrate for oxidative metabolism [17,88]. Or: ‘Fat-derived ATP production is designed to provide a “helper fuel” during exercise, with a maximum amount of energy at power outputs at ~60–65% VO2max [89,90].

As a result, the availability of carbohydrate ‘rather than fat, wins gold medals’ [91]. In contrast, ‘fat-rich diets do not improve training capacity or performance, but directly impair rates of muscle glycogenolysis and energy flux, limiting high intensity ATP production’ [80].

For the purposes of this review, the natural conclusion will be that since pre-exercise muscle glycogen content is determined by how much carbohydrate is ingested in the days prior to exercise (Figure 5), then a high-carbohydrate diet must always enhance performance in any exercise regardless of its duration or intensity. This is certainly the dietary advice that is now generally given [80,81,82,83,84,91,92,93,94].

However, as I have argued, these conclusions are entirely model dependent. They depend on a model that has not been proven and, as I have argued, is largely unprovable—that a limited capacity for carbohydrate oxidation secondary to restricted muscle glycogenolysis is the unique determinant of exercise termination. However, none of the iconic studies attempted to exclude the highly probable alternate explanation that, when it was measured, simultaneous hypoglycemia was a more likely cause of exercise termination (Figure 2, Figure 3 and Figure 4).

However, since some of those iconic studies have clearly established that reversal or prevention of hypoglycemia can reverse fatigue, at least temporarily in those whose muscle glycogen concentrations are also low [14,27,49], this possibility cannot be excluded forever, simply because it was never considered by the authors of the original iconic studies.

## 6. Why Can the Iconic Model of Exercise Fatigue That Developed from the Findings of the Original Iconic Scandinavian Studies Not Be Correct?

As described previously [6], there is one simple logical explanation for why this obligatory muscle glycogen/carbohydrate oxidation model cannot be the correct.

Already in 1987, twenty years after the iconic original studies of diet, exercise performance and muscle glycogen concentrations published by the Scandinavian researchers, R.K. Conlee [95] proposed that: ‘The most popular theory on why glycogen is necessary for energy production is the anaplerotic theory [96,97]. According to this theory, glycogen provides glucose moieties that are subsequently metabolized to pyruvate. The pyruvate is then carboxylated to oxaloacetate by pyruvate carboxylase. The net effect is a source of oxaloacetate to maintain citric acid cycle function. When glycogen runs out, citric acid cycle intermediates decrease, the processing of acetyl groups from beta oxidation is impaired, and the muscle fails from lack aerobic adenosine triphosphate (ATP) production. However, plausible and attractive as this theory is, it is unproven’.

‘Whether this is the major reason for glycogen depletion causing fatigue is not clear, but what is clear is that, in glycogen depleted muscle, ATP is being used up faster than it can be manufactured, and so force output is diminished’.

Conlee concludes his article with the following: ‘It is surprising that in spite of the considerable time that has elapse since these initial studies and the wealth of knowledge accumulated since, we still do not know the biochemical mechanism that explains why carbohydrate is essential to the ability of the muscle fiber to maintain high force outputs. It may be that the discoveries of 1967 were so widely accepted and so essentially unchallenged that we narrowed our vision as to the potential mechanisms responsible for the observations reported. The time has come for us to lay the issue to rest and answer the question: Why is glycogen availability necessary to perform prolonged, heavy intensity exercise?’

In answer to Dr Conlee’s question, what is absolutely certain is that the explanation he offers cannot be correct. For, if ATP is ‘being used up faster than can be manufactured’, then, as presented in Figure 12, the outcome must always be skeletal muscle rigor; not muscle fatigue.

Figure 12 shows that if the capacity for ATP production from muscle glycogen falls as it must during prolonged exercise [52], then the shortfall will have to be provided from the oxidation of alternate substrates including blood free fatty acids (mainly), liver-derived glucose, lactate and perhaps ketones. However, once the ATP demand exceeds the maximal capacity for energy production from all those fuel sources, then, according to Conlee’s explanation, the exercising skeletal muscles must develop a progressive ATP deficit. Additionally, if this is allowed to happen in an unregulated, non-homeostatic system, skeletal muscle ATP concentrations will continue to fall, ultimately leading to skeletal muscle rigor [98].

However, a number of studies have shown that skeletal muscle ATP concentrations are homeostatically regulated and well protected at fatigue.

For example, Fabbraio and Dancey [99] showed that muscle ATP concentrations were unchanged from resting values in 6 untrained subjects who exercised to exhaustion at ~65%VO2max (Figure 13). In addition, neither phosphocreatine (PCr) content nor the total adenine nucleotide pool (TAN = ATP + ADP + AMP), nor the degradation products inosine 5′-monophosphate (IMP) nor hypoxanthine were different from pre-exercise resting values.

The authors concluded that: ‘because TAN was not reduced, PCr was not depleted, and no correlation was observed between glycogen content and IMP when glycogen stores were compromised, fatigue may be related to processes other than those involved in muscle high-energy phosphagen metabolism’ [99].

One such possibility might be a developing hypoglycemia (Figure 2 in [99]) as blood glucose fell during exercise; this fall accelerated after 100 min of exercise, falling below 4.5 mmol/L thereafter.

Importantly, identical findings from the muscle biopsies were reported in a second similar study from this same laboratory [100].

Similarly, Parkin et al. [101] also reported that while muscle glycogen concentrations were very different at the termination of exercise in three different environmental conditions, muscle ATP (Figure 14) and TAN concentrations were not different from resting values measured before exercise.

The authors concluded that their results ‘demonstrate that fatigue during prolonged exercise in hot conditions is not related to carbohydrate availability’. Rather, they clearly show the action of a Central Governor regulating exercise in the heat [11] to ensure that subjects do not develop heatstroke.

Many others have expressed the similar opinion that muscle glycogen depletion causing a reduction in ATP concentrations in the exercising muscles is not the direct cause of fatigue. So, Fitts [33] concluded that ‘if glycogen depletion directly elicits fatigue by reducing energy production, cell ATP should decline and in general, this has not been observed’. Similarly, Hargreaves and Spriet [90] report that: ‘The muscle ATP concentration is reasonably well maintained, although it may decrease by ~20% during very intense exercise [102]’. This is because of ‘key regulatory mechanisms ensuring that ATP resynthesis is closely matched to the ATP demands of exercise’.

## 7. Two Other Studies Which Provide Definitive Disproof of the Obligatory Muscle Glycogen/Carbohydrate Oxidation Model

In his extensive review, Fitts [33] concluded that one (unproven) reason why high rates of carbohydrate oxidation may be ‘obligatory’ to sustain performance during prolonged exercise could be ‘to supply critical metabolic intermediate such that carbohydrate depletion reduces the oxidation rate of available fats and proteins’. There are two other studies in addition to those of Febbraio et al. [99] and Baldwin et al. [100] which disprove this obligatory muscle glycogen/carbohydrate oxidation hypothesis.

The main purpose of another study by Baldwin et al. [42] was to test whether ‘fatigue during prolonged exercise arises from insufficient intramuscular glycogen, which limits tricarboxylic acid cycle (TCA) activity due to reduced TCA cycle intermediates (TCAI)’. Thus, they wished to provide another test of the ‘anaplerotic theory’ of fatigue [95] which holds that only carbohydrate oxidation can maintain optimum ATP production in the exercising limbs especially during prolonged exercise

To test this hypothesis, the authors exercised 7 endurance-trained males to exhaustion at 70%VO2max when they began exercise with either high or low muscle glycogen concentrations. Muscle biopsies performed at the end of exercise were analyzed for TCAI concentrations.

Their results found that TCAI concentrations were not reduced at the termination of prolonged exercise when muscle glycogen concentrations were low. Thus, these data ‘do not support the concept that a decrease in muscle TCAI during prolonged exercise in humans compromises aerobic energy provision or is the cause of fatigue’ so that ‘our results do not support the hypothesis that links glycogen availability with fatigue via impaired aerobic energy provision’. Importantly, a progressive fall in blood glucose concentrations developed during exercise in both test conditions; the authors reported: ‘It is possible, however, that fatigue was associated with hypoglycemia and not intramuscular glycogen content, because glucose values declined to low levels in both trials…, values consistent with previous observations [42]’.

Their final conclusion was that: ‘Therefore, although glycogen availability influences exercise duration, our results refute the hypothesis that links attenuated glycogen availability with fatigue via impaired aerobic energy provision, impaired adenine nucleotide metabolism, or TCAI pool size’. In other words, the study found no evidence to support the obligatory carbohydrate oxidation theory. Instead there was some other unmeasured factor that explained fatigue.

The second study to investigate this hypothesis is that of Stellingwerff et al. [43]. The special importance of this study is that it also included a performance trial, missing in the study of Baldwin et al. [33]. However, for some undisclosed reason, the authors have never given the appropriate importance to the results of that performance trial.

Stellingwerff et al. [43] wished to determine any potential biochemical mechanisms to explain why adaptation to a high-fat diet reduces muscle glycogen use during subsequent exercise. In neither the abstract nor in the introduction to their paper do the authors make any mention of an attempt to link any ‘adverse’ metabolic adaptations to a possible impairment in exercise performance. Nor does the abstract include reference to the outcome of the performance trial.

For their study, seven well-trained male athletes exercised on two different occasions, once after eating a high-fat diet (67% fat; 18% carbohydrate; 15% protein) and once after eating a high-carbohydrate diet (15% fat; 70% carbohydrate; 15% protein).

The exercise involved 20 min of cycling at 70%VO2max followed immediately by a 1 min sprint at 150% of peak power output (PPO). After a 5 min rest, subjects performed a self-paced time trial at approximately 90% of VO2max for a total energy expenditure of 4 kJ/kg body mass. This exercise bout lasted approximately 10 min. Muscle biopsies were performed before; immediately after the start, and again at the end of the 20 min cycle at 70%VO2max; and again immediately after the end of the 1 min sprint at 150% PPO.

The muscle biopsy samples were analyzed to calculate (i) the contribution of glycogenolysis to energy production at the start of the exercise bout and after the 1 min sprint at 150% PPO; (ii) the substrate-level phosphorylation rate (which is the rate of ATP production from glycolysis and from PCr degradation); and (iii) the concentration of the active form of the enzyme pyruvate dehydrogenase (PDH) which regulates the conversion of pyruvate to acetyl-CoA for further metabolism in the tricarboxylic (Krebs) cycle. The concentration of this enzyme will influence the rate of production of tricarboxylic intermediates. Figure 15 shows the relevant results of the study.

Figure 15 shows that the calculated glycogenolytic rate was significantly less in the fat-adapted condition both at the start of the 70% VO2max exercise bout and at the end of the 1 min sprint at 150% PPO (panel A). However, the glycogenolytic rate increased 4-fold in the fat-adapted condition during exercise at 150% PPO compared to exercise at 70%VO2max. Thus, the glycogenolytic rate was significantly submaximal during exercise at 70% VO2max in both dietary conditions and was therefore not ‘limiting’ for exercise performance at 70% VO2max in either group. Rather, the data show clear evidence of homeostatic metabolic regulation with a reduced glycogenolytic rate during exercise of lower intensity, as is to be expected.

Although substrate phosphorylation (panel B) was lower in the fat-adapted condition during exercise at 70% VO2max, the 3–5-fold higher rates after exercise at 150% PPO were not different between dietary conditions. Hence, substrate phosphorylation could also not have ‘limited’ exercise at 70%VO2max after fat adaptation since very much higher values were achieved during all-out exercise which posed a much greater threat of metabolic failure. Instead, the lower value during exercise at 70%VO2max in both diet conditions is again evidence for metabolic homeostatic regulation, not metabolic ‘limitation’.

PDH activity (panel C) shows exactly the same results—values being substantially higher after the 150% PPO exercise than during exercise at 70%VO2max in both diet conditions. Thus, lower PDH activity could not have ‘limited’ performance in the fat-adapted condition since the values were submaximal and could have increased had there been an urgent metabolic need during exercise at 70%VO2max. Instead, the submaximal values during submaximal exercise provide further evidence for homeostatic metabolic regulation, not ‘limitation’.

Surprisingly, in neither the abstract nor the discussion did the authors mention the findings reported in panel D which show that exercise performance measured as either total time taken to complete the time trial; or mean power output; or power output as a %PPO was not different between diet conditions.

Thus, this study provides further definitive evidence disproving the anaplerotic theory. For, it shows (i) that none of the metabolic measurements ‘limited’ submaximal exercise performance since all increased substantially during all-out exercise and hence were not ‘limiting’ during exercise at a much lower exercise intensity (70%VO2max). However, more importantly, any small differences in these metabolic changes were not associated with differences in exercise performance which is ultimately the key outcome measurement in the exercise sciences.

Surprisingly, this paper has been cited frequently as strong evidence that the high-fat diet must impair exercise performance [103,104,105,106]. Yet the only appropriate title for the paper should have been: Depressed PDH activity and glycogenolysis during exercise following fat adaptation with carbohydrate restoration does not impair exercise performance.

## 8. Is Muscle ATP Utilization Regulated to Prevent ATP Depletion When Muscle Glycogen Concentrations Are Low? If So, What Is the Possible Mechanism?

The central governor model [6,11] predicts that it is not the rate of ATP resynthesis that ‘limits’ exercise performance. Rather, the ATP demands of exercise are the tightly regulated variable; regulation is achieved by controlling the number of motor units and hence muscle fibers that the motor cortex activates in the exercising limbs.

Indeed, Fitts [33] has proposed that perhaps muscle ATP concentrations are protected as a result of ‘fatigue produced by other factors reduces the ATP utilization before ATP becomes limiting’. A more recent review [107] also proposes that ‘traditionally this relationship (between muscle glycogen depletion and fatigue) has been attributed to a decreased ATP resynthesis rate due to inadequate substrate availability at the whole muscle level, but emerging evidence points to a direct coupling between muscle glycogen and steps in the excitation–contraction coupling including altered muscle excitability and calcium kinetics’. Others have suggested that this could represent the actions of a peripheral ‘governor’ located in skeletal muscle that regulates exercise performance [108,109].

However, this model is still ‘brainless’; it allows no role for the central nervous system [11], also to regulate exercise performance in order to avoid a ‘metabolic catastrophe’ [109].

Figure 16 shows that regulating the exercise capacity is the most reasonable solution to ensure that, should the capacity for ATP production be reduced, the cerebral motor cortex reduces motor drive to the muscles in the exercising limbs. This produces an immediate reduction in the capacity to do work. As a result, ATP demand falls instantantly, protecting muscle ATP concentrations.

There are a number of ways in which this reduction in exercise capacity could occur: The most probable explanation (in my opinion) is that regulation occurs as part of a feed-forward control mechanism from the brain and spinal cord, according to the central governor model [11,45]. There is a growing appreciation that this control might also be linked to glycogen metabolism by the brain astrocytes; since brain glycogen depletion would also have catastrophic effects [26,37,38]. Remarkably, like muscle, brain astrocytes increase their glycogen content during recovery from exhaustive exercise [26].

A dominant role of the central nervous system in regulating exercise performance to prevent a metabolic catastrophe, most certainly does not exclude an important and necessary contribution also from a ‘peripheral governor’ [108,109]. Experience teaches that the body never relies on a single mechanism for the homeostatic control of any specific variable; instead, the regulation is always complex—as it is for maintaining a normal blood glucose concentration [46].

However, is there any evidence that central brain control mechanisms regulate the exercise performance in an anticipatory manner as depicted in Figure 16?

Already in 2001, St Clair Gibson et al. [110] showed that power output during both the repetitive 1 and 4 km sprints during a 100 km self-paced time trial fell progressively (Figure 17). This fall could be due either to reduced central motor drive to the exercising muscles—control via a Central Governor—or as a result of peripheral fatigue through the action of a peripheral governor. If the effect is due to a peripheral governor, it is probable that an increased central feedforward output from the brain and spinal cord would attempt to compensate for any failure of power production by individual (fatiguing) muscle fibers. This would increase motor drive to those already fatiguing muscle fibers and also other, fresher muscle fibers in the exercising muscles. This effect would be detected as an increased electromyographic (EMG) activity in the exercising muscles.

However, the study found that EMG activity fell progressively during exercise (Figure 18), causing the resulting reduction in power output.

Thus, this finding is compatible with the interpretation presented in Figure 16; power output during prolonged exercise falls as a result of an anticipatory feedforward motor drive to the exercising muscles from the motor centers in the brain. This provides a homeostatic mechanism by which skeletal muscle ATP concentrations are always protected by a central control that reduces central motor drive whenever ATP demand threatens to exceed local ATP supply. As a result, muscle work falls, ATP demand falls and ATP homeostasis is retained.

The peripheral governor would provide a back-up safety regulator should the central regulation fail for any reason.

## 9. What Might Be the Effectors of This Feedforward Control Mechanism?

Our work on the central governor model suggested that the RPE plays a central role in exercise ‘regulation’ [111] since exercise seems always to terminate at high and maximal RPE values and that the rate at which the RPE rises during exercise can be used to predict the expected duration of that exercise bout [112]. As a result, the rate of rise in RPE can be used to predict the moment at which exercise will terminate [112]. In addition, the control acts ‘in anticipation’ [113,114,115]. Whilst researching this topic, I discovered the paper by Baldwin et al. [42], already discussed in part.

Unusually, Baldwin et al. [42] reported RPE values during exercise when athletes began exercise with either low or high muscle glycogen concentrations. Plotting those values against exercise duration showed that RPE rose as a linear function of both absolute and relative exercise durations [116] (Figure 19) as previously shown in other conditions [111,112].

Thus, I concluded: ‘One possible interpretation is that the RPE is the real determinant of the fatigue point in prolonged submaximal exercise at a fixed work rate. This is in keeping with the theory first proposed in the 1930s that exercise terminates when the feelings of discomfort overwhelm the potential rewards of continuing to exercise [117,118]. According to this interpretation, in the study of Baldwin et al. [42], the motivation to continue exercise stopped when a mean RPE of 18.1 units was reached in both conditions’.

‘These findings are compatible with a number of hypotheses, including the following. At the onset of exercise, on the basis of some carbohydrate signal that is increased after a carbohydrate-enriched diet, the subconscious brain calculates the anticipated duration of the exercise that can be safely sustained without causing absolute whole body energy depletion. Anticipating the maximal RPE that it (or the individual) will tolerate, the brain center responsible for the generation of the RPE then increases that RPE as a function of the percentage of that total exercise time that has been completed (or the percentage of time that remains). Alternatively, some signaling energy substrate, increased by a high-carbohydrate diet, may decline as a linear function of exercise duration, resulting in a linear increase in RPE. The maximum RPE is then reached before complete depletion of that energy substrate’.

‘Because similar RPE ratings at exhaustion occurred even though muscle glycogen contents were significantly different and were higher in the carbohydrate-replete trial, muscle glycogen content could not have been the exclusive determinant of the RPE in this study. Other sensory variables that also clearly did not contribute to this linear increase in RPE with increasing exercise duration were heart rate and oxygen consumption, which were little changed during exercise’.

‘The pivotal importance of the study by Baldwin et al. [42] is that it allows us finally to close the chapter on the concept that a state of absolute energy depletion is ever reached during prolonged exercise, just as it also does not occur during any other form of voluntary exercise [6]. The challenge now is to understand how the body anticipates the total duration of the exercise bout that is to be performed and how the increase in the perception of effort is regulated to ensure that skeletal muscle energy homeostasis is not sufficiently disturbed to produce local tissue damage, in particular, the development of skeletal muscle rigor’.

## 10. Published Studies Suggesting That Pre-Exercise ‘Carbohydrate Loading’ or Carbohydrate Ingestion during Exercise May Produce an Ergogenic Effect by Delaying the Time at Which Hypoglycemia Develops

The clear evidence is that when pre-exercise carbohydrate loading or carbohydrate ingestion during exercise improves performance during prolonged exercise to exhaustion, then the subjects in the control (non-carbohydrate condition) will develop a progressive hypoglycemia during exercise. Carbohydrate loading or carbohydrate ingestion that enhances performance in these studies always either prevents or reverses this hypoglycemia. This is so clearly established by 13 studies [14,17,27,49,51,57,58,59,69,70,71,72,73] that it appears to be a biological certainty. In contrast, if blood glucose concentrations remain in the normal range in the control condition, then carbohydrate ingestion [48,53,119] or pre-exercise carbohydrate loading [117] does not improve performance. The more relevant studies showing this effect of a falling blood glucose concentrations associated with a reduction in exercise intensity in the control condition are reviewed next.

(i)The study of Widrick et al. [71].

These authors evaluated the effects of carbohydrate ingestion on exercise performance of 8 subjects during four 70 km laboratory cycling time trials when athletes began exercise with either high or low muscle glycogen content (Figure 20). During exercise, subjects ingested either a 9% high-fructose corn syrup solution, or a sweetened placebo.

Figure 20 (panel A) shows that the cyclist’s power outputs were similar for the first 71% of all four trial. Thereafter, performance in the trial that subjects began with low pre-exercise muscle glycogen concentrations and did not ingest carbohydrate during exercise (low glycogen) fell progressively and was significantly less than in the other three groups. Importantly, only in that group did average blood glucose concentrations fall into the hypoglycaemic range (panel B).

Notable also was the finding that the performance of the cyclists when they began exercise with low muscle glycogen content but also ingested carbohydrate during the trial (low glycogen + CHO) was not different from their performance when they started exercise with high muscle glycogen content but did not ingest carbohydrate during exercise (high glycogen). Importantly, blood glucose concentrations at the end of those two trials were identical whereas muscle glycogen concentrations were much lower in the low glycogen + CHO trial (21.3 vs. 31.2 mmol/kg). Total muscle glycogen use was also approximately 40% greater in the high glycogen trial (139 vs. 85 mmol/kg).

The authors struggled to explain these findings according to the ‘brainless’ and ‘limitations’ models of obligatory carbohydrate oxidation to sustain prolonged exercise performance. Their problem was the rather too good performance of cyclists in the low glycogen + CHO trial compared to the high glycogen trial despite the substantially lesser use of muscle glycogen in the low glycogen + CHO trial. Those models cannot explain why performance was as good in the low glycogen + CHO trial when subjects used 40% less muscle glycogen during exercise and terminated exercise with much lower muscle glycogen concentrations than during the high glycogen trial.

They concluded that the decline in power output in the low glycogen trial resulted from ‘a reduction in the availability of muscle glycogen followed by, or in combination with, the failure of hepatic glucose production to maintain an adequate blood glucose concentration’ whereas ‘endogenous carbohydrate availability was sufficient under the high glycogen condition to enable subjects to maintain a relatively high rate of carbohydrate oxidation and to perform optimally throughout the time trial’.

Yet the overall differences in carbohydrate oxidation rates during the trials were seemingly trivial (3.0 g/min in the high glycogen trial vs. 2.4 g/min in the low glycogen trial). Indeed, the ingestion of 117 g of carbohydrate during the low glycogen + CHO trial increased the rate of carbohydrate oxidation by only 0.1 g/min to 2.5 g/min which was still somewhat less than the rates in the high glycogen trial (3.0 g/min).

In summary, the conclusions from this study appear clear. The ingestion of 117 g carbohydrate during ~120 min of exercise could not explain the differences in performance between the low glycogen + CHO and low glycogen trials according to the obligatory carbohydrate/muscle glycogen oxidation theory, since the effect on carbohydrate ingestion on the rate of carbohydrate oxidation was trivial (0.1 g/min).

However, the 117 g of ingested carbohydrate would be more than enough to prevent hypoglycemia, as it did (Figure 20) and as would be expected as the total blood glucose content is only 5 g [29,30].

(ii)The study of Flynn et al. [53].

This study somewhat confirmed the findings of the study of Wildrick et al. [71] as performances by 8 cyclists were not different during a 2 h exercise bout regardless of whether they ingested water or a total of 45–90 g of carbohydrates from one of four drinks with different carbohydrate concentrations or compositions.

Importantly, the subjects began all trials in a carbohydrate-loaded state (muscle glycogen concentrations > 180 mmol/kg) and their blood glucose concentrations remained within or above the normal range, even when they drank only water.

The authors concluded that ‘when experienced cyclists elevate muscle glycogen stores, the dependence on exogenous sources of carbohydrate during two hours of high intensity exercise is greatly reduced’.

However, they could also have concluded differently, specifically that provided subjects are able to maintain their blood glucose concentrations within the normal range when ingesting only water (placebo) during prolonged exercise, then carbohydrate ingestion during exercise will provide no benefit; essentially confirming the finding of Widrick et al. [71].

Another study [119], which failed to show any benefit of carbohydrate ingestion during a 100 km cycling time trial in a group of well-conditioned athletes, also found that blood glucose concentrations were well maintained for the entire duration of the exercise bout (Figure 2 in [119]).

(iii)The study of Mitchell et al. [51].

Mitchell et al. [51] exercised 10 trained cyclists for 2 h on 5 different occasions when they ingested either a water placebo or a carbohydrate solution with four different concentrations. The first 105 min of the trial was at 70%VO2max, the final 15 min was an all-out performance trial. Figure 21 shows changes in plasma glucose concentrations (panel A) and rates of carbohydrate oxidation (panel B).

Figure 21 shows that blood glucose concentrations remained within or above the normal range in all four trials in which carbohydrate was ingested. However, in the group ingesting water, blood glucose concentrations fell progressively dropping below 4.5 mmol/L after 75 min of exercise. Interestingly, rates of carbohydrate oxidation during the 15 min of all-out exercise appeared to be a function of the plasma glucose concentration after 100 min of exercise (Figure 21, panel B). However, performance during the 15 min of all-out exercise was only different between the group ingesting the most carbohydrate (CHO-18) and the water placebo group (WP).

Since total muscle glycogen use was not different in any of the groups, the authors concluded that ‘the improved performance may be the result of the maintenance of blood glucose (concentrations)’ and that exercise performance ‘may be sensitive to small changes in blood glucose levels’, which is predictable if the blood glucose concentration is the single most important metabolic variable that must be ‘protected’ [29,30,46].

They also noted that a carbohydrate dose of between 37 and 74 g was required to maintain or improve performance. Again, the important question is this: How does so little carbohydrate produce this effect without influencing either rates of carbohydrate oxidation during the first 105 min of exercise (panel B in Figure 21), or muscle glycogen use during the entire exercise bout?

(iv)The study of Williams et al. [72].

These authors [72] compared the performances of 18 runners, who, after completing a 30 km treadmill time trial, were randomly allocated to either a carbohydrate-supplemented group that increased their carbohydrate content for 7 days before they completed a second 30 km time trial, or to a control group that supplemented their diet with an increased consumption of fat and protein. Figure 22 compares the data for the group that supplemented their carbohydrate intakes before the second 30 km time trial.

Figure 22 shows that during their first 30 km race before carbohydrate supplementation, blood glucose concentrations began to fall after they had run 10 km (panel C); their running speeds began (panel A) to fall after 20 km when their blood glucose concentrations were approaching the hypoglycaemic range.

In contrast, after ‘carbohydrate loading’, blood glucose concentrations stayed above the starting value for the entire trial (panel D); running speeds were better maintained from 20 km to the finish (panel B). The group that did not ‘carbohydrate-load’ showed the same response in both 30 km time trials; their responses in both races (data not shown) mirrored those measured during the initial 30 km race in the group that subsequently ‘carbohydrate-loaded’.

‘Carbohydrate loading’ increased the rates of carbohydrate oxidation from 2.2 g/min in the initial trial to 2.5 g/min in the second trial. Thus, the ergogenic effect of carbohydrate loading was achieved through an increase in total carbohydrate oxidation during the second trial of just 33 g (330 vs. 297 g) expended over ~127 min of exercise. To achieve this effect, subjects had to increase their dietary carbohydrate intakes by 1138 g over 7 days.

The authors did not offer any specific explanations of how they thought the extra 33 g of carbohydrate oxidized after carbohydrate loading had improved the runners’ performances. Again, any such ergogenic effect can likely only be explained by the effect of 33 g in supplementing the 5 g normal found in blood [30] in order to prevent the development of hypoglycemia.

(v)The study of McConell et al. [57].

McConell and colleagues [57] exercised 8 endurance-trained cyclists to volitional exhaustion at 69%VO2max on two occasions, once when they ingested a sweetened placebo (CON) and the other when they ingested an 8% carbohydrate solution (CHO).

Exercise performance increased by 47 min (30%) when subjects ingested carbohydrate during the exercise bout (Figure 23). The total amount of carbohydrate ingested during the trial was 280 g.

Figure 23 (panel A) shows that blood glucose concentrations were elevated for the duration of the trial in the CHO trial but began to fall after 60 min in the CON trial reaching the hypoglycaemic range after 120 min. Muscle glycogen concentrations at exhaustion were higher in CON (panel B).

Muscle ATP, PCr and lactate concentration at the end of exercise were not different between trials but muscle IMP concentrations were significantly higher in the control condition.

The average carbohydrate oxidation rate collected ‘10–15 min before the onset of fatigue’ increased from 2.77 to 2.88 g/min as a result of carbohydrate ingestion. However, importantly, this was not different between trials.

Since skeletal muscle IMP concentrations were increased at exhaustion in the control condition, the authors concluded that: ‘These results imply that carbohydrate ingestion may increase work capacity during prolonged exercise, at least in part, by improving metabolic energy supply within contracting skeletal muscle’.

However, to arrive at this conclusion, the authors had to ignore the very marked differences in plasma glucose concentrations between the trials and especially at the point of exhaustion, as well as the trivial difference in rates of carbohydrate oxidation in the two dietary conditions.

In short, this study provided no evidence that subjects terminated exercise because of a failure of an obligatory carbohydrate oxidation necessary to sustain prolonged exercise performance. Rather, a progressive hypoglycemia develops in CON and was prevented by carbohydrate oxidation in the CHO trial.

(vi)The study of Angus et al. [49].

These authors had 8 trained cyclists complete a 100 km cycling time trial as quickly as possible whilst ingesting either a sweetened placebo (P), a 6% carbohydrate solution (C), or a 6% carbohydrate solution with added medium chain triglycerides (C + M). Time to complete the trial was significantly slower when P was ingested compared to C and C + M ingestion (Figure 24).

Figure 24 (panel A) shows that, when carbohydrate was not ingested, the initial metabolic change already present after 30 min of prolonged exercise was a steep fall in plasma insulin concentrations; this allowed an increase in circulating free fatty acid concentrations and in fat oxidation (not shown). Plasma glucose concentrations (panel B) peaked after 30 min in the P trial, falling almost linearly thereafter until the end of exercise when hypoglycaemic levels were reached.

In contrast, normal blood glucose concentrations were maintained by carbohydrate ingestion in C and C + M.

The average work rate (panel C) fell linearly and irreversibly, beginning after 105 min only in the P condition. The fall mirrored the fall in plasma glucose concentrations in that condition. The rate of carbohydrate oxidation (panel D) was maintained when carbohydrate was ingested but fell progressively and linearly beginning after 90 min in the P condition, also mirroring the change in plasma glucose concentrations and in the average work rate.

Whilst noting the marked reduction in blood glucose concentrations in the control experiment, the authors concluded only that: ‘It was only in the final segment of the time trial that a reduction in carbohydrate oxidation, RER, and plasma glucose combined to reduce the amount of work performed if exogenous carbohydrate was not provided’.

However, this conclusion is true only according to the ‘metabolic limitations’ model which holds that there is an obligatory requirement for muscle glycogen oxidation during prolonged exercise.

According to the alternate homeostatic control model which posits that the blood glucose concentration is the protected variable so that the exercise intensify is regulated ‘in anticipation’ to ensure that profound hypoglycemia does not develop, their explanation describes reverse causation.

Rather, it is far more likely that the falling blood glucose concentration (panel B) causes a controlled reduction in the average work rate (panel C) as the result of a centrally-directed reflex (missing in the original model shown in Figure 9) resulting in a protective reduction in carbohydrate oxidation (panel D) so that the blood glucose concentration is ‘protected’.

It is possible that feedforward from those brain control centers might, in response to a progressive hypoglycemia, also reduce blood glucose oxidation by the exercising muscles (Figure 9).

(vii)The study of Febbraio et al. [70].

In a related study from the same laboratory [70], performance during a time trial that followed 120 min of exercise at 63% VO2max was clearly determined by the blood glucose concentrations at the end of 120 min of exercise. This explains why the authors concluded that: “pre-exercise ingestion of carbohydrate improves performance only when carbohydrate ingestion is maintained throughout exercise”. Only regular glucose ingestion for the duration of the exercise bout was sufficient to maintain blood glucose concentrations in the normal range (Figure 1 in [70]), allowing subjects to begin the time trial with normal or elevated blood glucose concentrations.

(viii)The placebo-controlled carbohydrate-loading study of Burke and colleagues.

Professor Louise Burke and her colleagues [120] performed what was perhaps the first placebo-controlled study of the effects of carbohydrate loading on subsequent exercise performance. It remains one of few such placebo-controlled studies.

The original iconic study [19] involved athletes who knew when they were carbohydrate-loaded and when they were not. The scientists conducting those research studies would also have known when their athletes were ‘carbohydrate-loaded’. The absence of blinding must always influence the study outcome. The findings of the placebo-controlled study of Burke et al. are shown in Figure 25.

Figure 25 shows that performance during repetitive 4 km (panels A and B) or 1 km (panels C and D) sprints during a self-paced 100 km time-trial were not different when athletes ‘carbohydrate-loaded’ with either a placebo or with real carbohydrates. Time to complete the overall 100 km time trial was also not significantly different between conditions.

Thus, this study failed to find any benefit of carbohydrate loading when the subjects (i) performed stochastic exercise; (ii) were unaware whether they had received either the placebo or the carbohydrate intervention; (iii) when they did not fast for 12–16 h before exercise and (iv) when they also ingested carbohydrate during exercise. Although blood glucose concentrations were not measured during the trial, it is a fair assumption, based on the published literature, that these did not fall during exercise.

The study therefore disproves the idea that carbohydrate loading with a high-carbohydrate diet must improve performance in subjects who are also ingesting carbohydrate during exercise. So, the authors, including myself, wrote the following: ‘In summary, this study shows that a CHO-loading protocol that increases pre-exercise muscle glycogen concentrations resulted in a minimal effect on the performance of a 100 km time trial involving high intensity sprints when CHO was ingested before and during the event according to contemporary sports nutrition guidelines. Although we cannot completely discount the possibility that CHO loading has a worthwhile effect on the performance of competitive cyclists… our data suggest that this effect is small…Furthermore, this study raises the possibility that part or all of the ergogenic effect of CHO loading reported in previous studies could result either from a placebo effect or from higher pre-exercise liver glycogen stores that could delay the onset of hypoglycemia during prolonged exercise’ [120].

This conclusion is compatible with all the evidence presented here that the main performance-enhancing effect of ‘carbohydrate loading’ may be to increase pre-exercise liver glycogen stores and so to delay the onset of fatigue-inducing hypoglycemia during prolonged exercise.

If this is so, then the proper model of human exercise performance needs to include the role of the central nervous system as a homeostatic regulator [11]; in place of the more simplistic model which concludes that exercise performance will be impaired whenever the capacity for skeletal muscle carbohydrate oxidation is reduced, for example, by ‘limiting’ muscle glycogen stores.

## 11. Other Studies of Metabolic Interventions to Identify Possible Mechanisms Explaining Fatigue during Prolonged Exercise

(ix)Studies of metabolic inhibition with nicotinic acid.

Two or the original Scandinavian researchers, Drs. Pernow and Saltin [121] completed a small study on four subjects who exercised after ingesting nicotinic acid which inhibits free fatty acid release from adipose tissue, thereby reducing fat oxidation. They found that nicotinic acid impaired performance in athletes who began exercise with low starting muscle glycogen concentrations so that “when the glycogen stores are reduced, prolonged exercise can still be performed on submaximal levels (less than 60–70% of the maximal oxygen uptake) provided that the supply of free fatty acids is adequate. Elimination of both muscle glycogen and exogenous free fatty acid seriously impairs the ability for prolonged exercise’. However, this conclusion is complicated by the presence of profound hypoglycemia during the nicotinic acid trials in three of the four subjects.

A subsequent more detailed study by the same researchers [122] found no effect of nicotinic acid on performance. Blood glucose concentrations remained in the normal range in those studies.

A series of more recent studies [83,88] also found no effect of nicotinic acid on exercise performance when the exercise bout was less than 90 min. In one study, exercise performances during time trials of 30, 60, 90 and 120 was impaired only in the longest exercise test. Additionally, only in that longest trial did blood glucose concentrations fall in the final 30 min of exercise (Figure 3C in [88]) at the same time that the work rate was significantly lower following nicotinic acid ingestion.

These studies of metabolic inhibition of circulating free fatty acids with nicotinic acid suggest that the negative effects of performance are most likely related to an increased reliance on liver glycogen use during exercise, leading to more rapid hypoglycemia. Thus, the ‘carbohydrate dependence’ in these studies appears to be for carbohydrates to maintain blood glucose concentrations.

(x)Studies of cerebral glucose metabolism during exercise-induced hypoglycemia; including the identification of a feedforward control that limits exercise performance during hypoglycemia.

Nybo and colleagues [123] studied cerebral blood flow and brain metabolism in 6 endurance-trained athletes during three hours of exercise at a fixed work rate of ~210 W when they ingested either 200 g glucose or placebo (Figure 26).

Figure 26 (panel A) shows that arterial glucose concentrations fell progressively in the placebo group, dropping below 3 mmol/L at the end of 180 min of exercise. The arteriovenous (A-V) difference for glucose across the brain fell steeply after 120 min of exercise (panel B)—indicating a reduced cerebral glucose metabolism. This remained suppressed for the final hour of exercise. Ratings of perceived exertion (panel C) rose steeply after 120 min, at the time when blood glucose concentrations fell below ~3.5 mmol/L and the A-V glucose difference fell sharply (panel B).

Although cerebral blood flow was the same in both trials, cerebral glucose uptake, cerebral oxygen consumption and cerebral metabolic rate all fell when hypoglycemia developed in the placebo trial.

Thus, hypoglycemia reduced glucose uptake by the brain, impaired cerebral metabolism and caused a progressive rise in the RPE. It can be predicted that had exercise continued in the placebo trial, it would have terminated within approximately a further 15 min as the RPE increased above 18.

Nybo [124] also measured knee extensor force in the same subjects within approximately 20 s of their completing the 180 min exercise bout during which they ingested either carbohydrate or placebo. Subjects performed a sustained 120 s maximal quadriceps contraction during which brief electrical stimulations were superimposed every 30 s, beginning at 30 s.

When compared to the force generated voluntarily, the knee extensor force produced by direct electrical stimulation provides a measure of the degree to which the motor units in the exercising limb muscles are activated by voluntary exercise in the tested individuals. Results were compared between the two trials and with those measured before the exercise began.

The results showed that voluntarily developed knee extensor force was lower in both trials compared to values measured before exercise. Importantly, force development was the least after the placebo trial. This was due to significantly reduced central motor activation by the motor cortex at the termination of placebo trial when subjects were profoundly hypoglycaemic (panel A in Figure 26).

Nybo [124] concluded that: “exercise-induced hypoglycaemia in endurance-trained subjects lowers the average force production during a sustained maximum muscle contraction, and the reduced force development is associated with the diminish activation drive from the central nervous system”.

Hence, the authors provide the evidence for the feedforward control required to prevent hypoglycaemic brain damage during prolonged exercise when carbohydrate is not ingested. Addition of this control mechanism completes the original diagram from which Figure 9 originates, which can now explain how this homeostatic mechanism work seffectively also during prolonged exercise.

However, Nybo’s [124] explanation continues to promote a ‘limitations’ model, as do others [125], in which hypoglycemia is considered to cause ‘central fatigue’. Rather than to be the central component of an exquisite control mechanism, evolved over eons in mammals, to ensure blood glucose homeostasis especially during prolonged exercise when carbohydrate is not ingested.

(xi)The studies of Weltan et al.

The studies of Weltan et al. [64,65] began as an offshoot of our attempt to understand why exercise performance is impaired in those who begin exercise in a carbohydrate-depleted state [126]. That study [126] found that carbohydrate infusion increased blood glucose concentrations and rates of glucose oxidation and extended time to fatigue in subjects who began exercise in a carbohydrate-depleted state. It also established that the response to glucose infusion was highly individualized.

This is to be expected since the response of cerebral metabolism to a developing hypoglycemia will be highly dependent on the extent to which the subject’s brain is adapted for fat oxidation and, in particular, to the use of ketone bodies [127,128,129]. When fully adapted for example, to prolonged starvation, ketone bodies can provide up to 67% of the brain’s energy requirement, compared to 0% in subjects eating high-carbohydrate diets [127,128,129].

That the brain shows this ‘metabolic flexibility’ only when adapted to starvation or to a high-fat diet and that it lacks such flexibility when exposed to a high-carbohydrate diet has generally been ignored in discussions of what diet is optimum for performance during prolonged exercise.

Indeed, the study of Nybo et al. [123] showed that the brain adapted to a high-carbohydrate diet lacks this flexibility and is unable acutely to substitute energy derived from ketone body metabolism to make up for any deficit caused by a reduced cerebral glucose uptake in acute hypoglycemia. Hence, athletes who are fat-adapted would be less likely to exhibit hypoglycaemic symptoms during prolonged exercise and also less likely to show benefit from glucose ingestion or infusion in experiments similar to those undertaken by Claassens et al. [126].

However, in the study of Claassens et al. [126], fatigue ultimately developed even in subjects whose blood glucose concentrations were maintained in the normal range by glucose infusion. Thus, these findings are identical to hose reported by Cogan and Coyle [27] (Figure 8) and Coyle et al. [49] (Figure 10).

The studies of Weltan et al. [65,66] have established that the increase in fat oxidation present in those who begin exercising with depleted muscle (and liver) glycogen stores and resulting low blood insulin concentrations, is not reversed by glucose infusions sufficient to produce hyperglycemia. Instead, the data show that the rate of both muscle glycogen use as well as the overall rates of carbohydrate and fat oxidation during exercise are established, before exercise, by the size of the pre-exercise muscle and very probably liver glycogen stores [65]. Perhaps it is these depleted stores that alter the insulin response to ingested or infused glucose; instead maintaining low blood insulin concentrations during exercise even though the glucose infusions in these experiments produced a profound hyperglycemia.

These studies reveal that the clear goal of the homeostatic mechanisms activated during prolonged exercise that lowers liver and muscle glycogen stores are to maximize rates of fat oxidation in order to constrain, and not to maximize carbohydrate oxidation, especially as muscle and liver glycogen concentrations fall progressively. A key effect will be to reduce the rate of blood glucose oxidation in order to delay the development of hypoglycemia.

As the authors explained: ‘Thus glucose oxidation is not (current author’s emphasis) increased by reduced muscle glycogen content in subjects with similar blood glucose concentrations; instead, a switch takes place towards lipid oxidation even when plasma glucose concentrations are hyperglycemic. This strengthens the argument in our previous study [64] that this may be a teleological mechanism to compensate for a reduced availability of intramuscular carbohydrate availability without predisposing to hypoglycaemia (current author’s emphasis)’.

In other words, the goal of the homeostatic control mechanisms revealed by these studies is to protect against liver glycogen depletion leading to hypoglycemia, and not to slow the rate of muscle glycogen use to protect against muscle glycogen depletion. The evidence is clear that neither glucose infusion (Figure 11) nor glucose ingestion reduces the rate of muscle glycogen use during exercise.

These data also show that the belief that carbohydrates are the preferred fuel during prolonged exercise [80,83,88] is not universally true. Rather it depends on the prevailing blood insulin concentrations which are a function of the starting muscle and liver glycogen concentrations.

Thus, when faced with the choice of burning fat or carbohydrate when carbohydrate is provided in excess in the carbohydrate-depleted state, the body clearly chooses to use fat over carbohydrate ([65,66] Figure 8 in [65]) in part by maintaining lower blood insulin concentrations with resulting higher circulating free fatty acid concentrations and reduced blood glucose oxidation.

The extent to which this effect is regulated by the liver rather than muscle glycogen stores is currently unknown. However, the point is clear: when the body senses it is in a carbohydrate-depleted state, it moves to maximize fat and limit carbohydrate oxidation. The key outcome is that this will reduce blood glucose oxidation so as to prevent or delay the onset of hypoglycemia.

The reasons why hypoglycemia must be prevented at all costs during exercise, at least in those whose brains are not adapted to ketone metabolism, should, by now, be apparent.

Summary:

Summarizing the findings of these studies, a reasonable explanation might be the following: when carbohydrate is not ingested during prolonged exercise, the metabolic response is to minimize carbohydrate oxidation in order to prevent a potentially fatal hypoglycemia. The hypothalamic brain homeostatic mechanisms achieve this by immediately reducing blood insulin concentrations in order to increase rates of fat oxidation; if this fails to prevent a progressive hypoglycemia, these same mechanisms will then begin to restrain the amount of work it will allow the exercising muscles to perform. If this fails, ultimately the athlete will collapse with a paralyzing hypoglycemia as the brain reduces the allowed work output to zero, thereby preventing brain damage from profound hypoglycemia.

This in turn means that if the nutritional goal is to improve endurance performance by slowing muscle glycogen use during exercise, then some mechanism other than pre-exercise carbohydrate loading or carbohydrate ingestion during exercise will need to be developed and tested. At present, the only known method likely to reduce muscle glycogen use during exercise is to pre-adapt to a high-fat diet [130,131,132,133]. In one study, the rate of muscle glycogen use during 2 h of exercise at 72%VO2max was 50% lower in chronically fat-adapted than in carbohydrate-adapted athletes [133].

However, at present, the more obvious approach would be to slow blood glucose use in order to delay the development of hypoglycemia.

## 12. Results from a Series of Meticulously Conducted Studies Which Suggest That a High-Fat Diet Is Not Appropriate for Competitive Endurance Athletes

(i)The study of Havemann et al.

By failing to consider the role of the central governor to explain the findings in a study from my former laboratory [104] and of which I was a co-author, we invited a carefully argued editorial describing that the results of that study provided the final “Nail in the Coffin” [103] for the idea that a high-fat diet might provide value for competitive endurance athletes.

This study used the same protocols as those developed by Burke et al. [120] and St Clair Gibson et al. [110]—stochastic exercise in which 1 and 4 km sprints were included at regular intervals during a self-paced 100 km time trial. However, unlike the findings in the study of Burke et al. [120] (Figure 25), specifically that carbohydrate loading provided no benefit over a placebo control, in this study there was clear evidence for an apparent performance impairment when athletes ate a high-fat diet for six days before the time trial (Figure 27).

Figure 27 shows that exercise performance in the second, third and fourth 1 km sprints was significantly impaired following low-carbohydrate high-fat diet, even though neither overall finishing time for the 100 km time trial nor performance in the 4 km sprints was impaired.

The authors, myself included, concluded: ‘...(the high-fat followed by carbohydrate) dietary strategy increased fat oxidation, but compromised high intensity sprint performance, possibly by increased sympathetic activation or altered contractile function’. Although we did note that: ‘There was an increase in power output during the final 1 km sprint in both…trials, which is indicative of a reserve capacity’ we erroneously chose to interpret the findings according to the obligatory carbohydrate oxidation model of exercise limitation rather than according to the central governor model.

In their editorial [103], Professors Burke and Kiens made the reasonable proposal that the study provided definitive evidence that the low-carbohydrate diet ‘compromises the ability of well-trained cyclists to performance (sic) high intensity sprints. As a result ‘…it seems that we are near to closing the door on one application of this dietary protocol. Scientists may remain interested in the body’s response to different dietary stimuli and may hunt for the mechanisms that underpin the observed changes in metabolism and function. However those at the coal-face of sports nutrition can delete fat loading and high-fat diet from their list of genuine ergogenic aids for conventional endurance and ultra-endurance sports’ [103].

However, on closer examination, it is clear that the impaired performance during the second, third and fourth 1 km sprints in the study (Figure 27) could not be due to failure of obligatory carbohydrate oxidation in the second, third and fourth 1 km sprints that occurred at 32, 52 and 72 km respectively. For the reason that athletes ‘recovered’ their full power outputs in the final 1 km sprint at 99 km and which occurred after a further 27 km (27%) of the 100 km time trial. If their impaired performance in the earlier sprints had been caused by a failure of obligatory carbohydrate oxidation at 32, 52 and 72 km, then this effect would have been even more exaggerated in the final sprint at 99 km. In addition a proposed failure of obligatory carbohydrate oxidation did not influence 4 km sprint performances at 40, 60 and 80 km; nor overall performance from 32 km to the finish.

According to the central governor model, in order to complete the 100 km time trial, subjects naïve to the biological effects of the low-carbohydrate diet, chose an altered pacing strategy, conserving their effort for a classic end-spurt in the final 1 km sprint (Figure 27).

In summary, the study used to ‘delete’ high-fat diets from consideration could only be interpreted as the final “Nail in the Coffin” if one considers only the as yet unproven [42,43] model in which high rates of carbohydrate/muscle glycogen oxidation are obligatory for the sustained performance of prolonged exercise.

Importantly, in the study of Havemann et al. [104], blood glucose concentrations were the same during both trials following the two different dietary interventions. Thus, any difference in performance could not be explained by the presence of hypoglycemia following the low-carbohydrate diet.

(ii)Three exceptional studies appear to establish that the performance of Olympic-quality 50 km race walkers is significantly impaired when they follow a low-carbohydrate high-fat diet for 3 weeks.

Over the past 7 years, Professor Louise Burke and her team have conducted a series [134,135,136] of quite remarkable studies of the effects of different dietary interventions on Olympic race walkers. The studies are exceptional because they involve world-class athletes and have provided a wealth of unique metabolic data that are of the greatest value. However, for five reasons they also do not provide the definitive ‘Nail in the Coffin’ evidence for or against the value of different diets on exercise performance. The studies are therefore hypothesis generating; they are not hypothesis testing.

First, elite athletes preparing for Olympic competition cannot be expected to commit to randomization to a novel diet, that is, to participate in a randomized controlled trial (RCT). The authors explained: “The design and implementation of the study involved a pragmatic blend of rigorous scientific control and research methodology with real-world allowances needed to accommodate elite athlete populations” [134].

Thus: “Prior to their arrival to the study camps, participants were educated about the benefits and limitations of the different dietary treatments and asked to nominate their preference(s) for, or non-acceptance of, each of these interventions” [134].

Explaining ‘benefits and limitations’ of the different dietary treatments will introduce bias either for or against specific dietary interventions, removing the possibility of randomization. Some of the author’s opinions about the relative values of the different dietary interventions have been repeatedly expressed in the scientific and lay literature and would very likely have been known to some or all of the study participants.

Second, the athletes could not be ‘blinded’ to the dietary intervention they would receive since each had personally chosen the intervention he would receive. The researchers in charge of the exercise testing may also have known the nature of the diet followed by the different subjects.

Third, the period of adaptation to the diet was quite short—less than 4 weeks. A number of elite athletes have expressed the opinion (to me) that adaptation to the low-carbohydrate high-fat diet usually takes longer than 3–4 weeks.

Fourth, subjects in the intervention group were exposed to high-intensity training whilst also radically altering their diets. Thus, the trial examined the effects of a radical dietary change during a period of highly stressful training.

Fifth, subjects in the low-carbohydrate group did not receive supplementary carbohydrate during the conduct of the 25 km time trials.

These considerations, particularly that none of these studies was an RCT, negate definitive conclusions about the performance effects of the LCHF diet in this trial. Nevertheless, the study produced some profoundly interesting findings.

In the first place, athletes developed remarkably high rates of fat oxidation during exercise; this adaptation was present even within 5–6 days of beginning the low-carbohydrate diet [136]. For example, Figure 28 shows that rates of carbohydrate and fat oxidation were dramatically altered by exposure to the high-fat diet for 3–4 weeks; similar results were already measured within 5–6 days [136].

Thus, Figure 28 shows that during the final 25 km time trial at the end of the 3–4 week experiment, subjects eating the low-carbohydrate high-fat diet had increased their rates of fat oxidation to >1.5 g/min compared to rates of ~0.5 g/min prior to fat adaptation. Fat adaption reduced carbohydrate oxidation rates to below 0.5 g/min, compared to rates of ~3 g/min in athletes following either of the two high-carbohydrate diets.

Importantly, subjects in the low-carbohydrate group performed, on average, worse in all performance trials in all three studies; in two of these studies (Figure 6A in [134]; Figure 5A in [136]) blood glucose concentrations were lower in the low-carbohydrate group during the 25 km time trial. In the third trial blood glucose concentrations were not measured [135].

Importantly, race walkers eating the higher carbohydrate diets received supplementary carbohydrates during the trial in the form of sports drinks and sports gels for an average intake of 60 g/hr. In contrast, the athletes in the low-carbohydrate group received only water and ‘low-carbohydrate high-fat cookies’ [134]. Their carbohydrate intakes during these 25 km time trials were not reported.

Differences in carbohydrate intake during the 25 km races might explain differences in blood glucose responses between groups. With the result that, according to all the evidence presented here, the failure to provide the race walkers adapting to the low-carbohydrate high-fat diet with sufficient carbohydrate during the 25 km time trials might be sufficient to explain their mildly impaired performances during those trials.

Interestingly, when the trial was a real competition—the 20 km Australian Race Walking Championships—7 of the 19 subjects in one trial produced life-time personal best times [135]. Of these seven athletes, four were in the high-carbohydrate and three in the low-carbohydrate group. Thus, 4 of 11 (36%) athletes in the high-carbohydrate group improved their performance on the high-carbohydrate diet compared to 3 of 8 (38%) of those athletes who radically altered their diets to reduce their carbohydrate intake. This suggests some degree of parity between the likelihood that athletes would benefit from the two different dietary interventions.

The importance of this finding is to disprove any claim that the low-carbohydrate high-fat diet produces an irreversible metabolic ‘limitation’ which prevents world-class athletic performances in athletes eating low-carbohydrate diets on the grounds that such world-class performances are ‘carbohydrate dependent’.

Rather, this finding proves that diet is just one of many factors that influences individual athletic performance. Additionally, it may not be the most important.

Interestingly, one world-class race walker, not involved in those studies, self-chose to adopt the low-carbohydrate high-fat diet in the spring of 2014 [137]. Thereafter, he became both the 2015 World 50 km Race Walking Champion and followed this up with victory in the 2016 Olympic 50 km race. He explained: “I was concerned about my energy crises at 35 km mark (when eating the high carbohydrate diet). My error was that I became too dependent on carbohydrates. We trained the body, (and) muscles, not metabolism” [138].

These studies emphasize that laboratory studies produce results that may not be generalizable to every single athlete across the globe. Perhaps the message is that athletes need to read the scientific literature. However, then they need also to test the predictions of the scientists on themselves.

Finally, the authors generally use the argument that the loss of performance is due to a failure of carbohydrate oxidation induced by the diet. However, the reality is that the metabolic adaption to the diet will be homeostatically controlled as are all such adaptations in the human body.

Why would the body choose to impair its exercise capacity by reducing rates of carbohydrate oxidation to levels (panel A in Figure 28) that apparently cannot sustain peak exercise performance (according to this explanation)?

Furthermore, why did this restraint of carbohydrate oxidation yet allow 38% of athletes in one study [135] to achieve life-time best performances?

Human exercise performance is complex and should not be reduced to considerations of a single variable [11,44,45].

## 13. The Review of Karelis et al.

Karelis et al. [139] have presented a review of the possible mechanisms by which carbohydrate administration might enhance athletic performance. They suggest the following six mechanisms could be involved:(i)An attenuation of central fatigue.(ii)Better maintenance of carbohydrate oxidation rates.(iii)Muscle glycogen sparing.(iv)Alterations in muscle metabolite levels.(v)Reduced exercise-induced strain.(vi)Improved maintenance of excitation–contraction coupling.

They conclude that, as clearly reviewed here (Figure 10, Figure 11 and Figure 23), carbohydrate administration does not reduce muscle glycogen use during exercise. Instead, it reduces the reliance on blood glucose oxidation and hence protects against liver glycogen depletion and the premature development of hypoglycemia.

They also find no evidence that there is a dose-dependent relationship between the amount of carbohydrate ingested during exercise and any increase in exercise performance (Figure 29). This might be expected if a relatively small carbohydrate intake is sufficient to prevent the development of hypoglycemia.

Thus, Figure 29 shows that there is no effect of an increasing carbohydrate intake on performance during exercise. Since more carbohydrate ingested during exercise increases the rate of carbohydrate oxidation, this finding suggests that, if carbohydrate ingestion during exercise improves athletic performance, it is not the result of an increased rate of carbohydrate oxidation.

Instead, the authors suggest that: ‘Emerging evidence from the literature shows that increasing neural drive and attenuating central fatigue may play an important role in increasing performance during exercise with carbohydrate supplementation’.

I would argue that the emerging evidence is that exercise is homeostatically regulated, not ‘limited’, specifically to ensure that hypoglycaemic brain damage does not occur.

This is the first priority.

However, it is also clear that prolonged exercise cannot continue indefinitely even in those provided with an external source of carbohydrate at the point of exhaustion [14,27,49,126] and who are not hypoglycaemic when they subsequently terminate exercise (Figure 2, Figure 3, Figure 8 and Figure 10).

Thus, some other as yet unidentified regulators exist as the ultimate terminators (via feedback to the central nervous system) of prolonged exercise performance.

Furthermore, these final ‘terminators’ may not be influenced by the dietary macronutrient choices of the athletes.

## 14. The Complete Analysis of 28 Randomized Controlled Trials of High- or Low-Carbohydrate Diets on Exercise Performance

If it is true that the low-fat high-carbohydrate diet produces a sub-optimal metabolic state that ‘limits’ exercise performance, then this evidence must be obvious from randomized controlled clinical trials comparing the effects of high-carbohydrate versus low-carbohydrate high-fat diets on athletic performance when athletes are fully adapted to either state.

To date, at least 28 published studies [130,131,140,141,142,143,144,145,146,147,148,149,150,151,152,153,154,155,156,157,158,159,160,161,162,163,164] have now unequivocally established [165,166] that in the vast majority of athletes, the two different diets produce *equivalent* performances across a wide range of athletic endeavors.

Indeed, two recent meta-analyses conclude that: “Overall, the majority of null results across studies suggest that a (low carbohydrate) ketogenic diet does not have a positive or negative effect on physical performance compared with a control diet” [165] and “…moderate to vigorous intensity exercise experiences no decrement following adaptation to a ketogenic diet” [166].

Thus, the evidence at present is clear. Chronic adaptation to either a high-carbohydrate or a low-carbohydrate high-fat diet will produce similar outcomes in terms of athletic performance.

This might suggest that contrary to the popular interpretation of the iconic Scandinavian studies [23,24], perhaps in the final analysis, the specific macronutrient content of the diet has very little impact on exercise performance.

Perhaps the athlete’s body has an extraordinary capacity to adapt to whatever diet the experts promote as the optimum.

## 15. The Potential Negative Long-Term Health Consequences of Promoting High-Carbohydrate Diets for Athletes Who Are Insulin Resistant

The rising prevalence of obesity and type 2 diabetes mellitus (T2DM) around the world has begun to focus attention on the underlying metabolic state for both conditions, and indeed of most chronic modern diseases, which is insulin resistance. It is estimated that insulin resistance may be present in as many as 80% of adult North Americans [167].

It is also clear that whilst exercise may protect against some aspects of insulin resistance, it is unable to prevent its remorseless progression to T2DM in those who continue to eat diets that produce persistently raised blood insulin concentrations (hyperinsulinemia) over many decades.

The important point is that even elite athletes eating high-carbohydrate diets will not be immune from these detrimental effects if they are insulin resistant. Two recent studies have drawn attention to this phenomenon.

With the use of continuous blood glucose monitoring, Flockhart et al. [168] have shown that (some) world-class endurance athletes have impaired glucose control compared to a matched control group. Blood glucose concentrations were in the hyperglycaemic range (>8 mmol/L) for nearly twice as long (41 min/24 h) in these athletes compared to a control group of non-athletes. Hyperglycemia typically occurred at between 1 and 2 pm each day (Figure 7D in [168]) suggesting a dietary influence.

Also using continuous glucose monitoring in 10 sub-elite athletes, Thomas et al. [169] found that 4 of the 10 subjects spent more than ~70% of the total monitoring time with blood glucose concentrations greater than 6.0 mmol/L indicating pre-diabetes.

The authors concluded that: ‘Athletes are traditionally encouraged to consume high carbohydrate diets to replenish muscle glycogen stores and improve performance, with a particular focus on postexercise carbohydrate consumption [170,171,172]. However, this advice may be negatively impacting the blood sugar levels of athletes predisposed to have a low tolerance of carbohydrates. …Hence, the potential for a more personalized nutrition plan aided by continuous glucose monitoring to optimize the blood glucose levels during different phases of athletes training is highlighted by these results’ [169].

Importantly, a key driver of insulin resistance is the presence of visceral obesity and non-alcoholic fatty liver disease (NAFLD) [173] that is caused by sugar containing high-carbohydrate diets in those who are already insulin resistant [174].

Interestingly, the high-fat low-carbohydrate diet is the most effective method to reduce visceral obesity and NAFLD [175,176,177], whereas diets high in fructose including high-fructose corn syrup (HFCS) and or sugar or both promote NAFLD and visceral obesity [178,179]. Typically, so-called ‘sports drinks’ and many other high-carbohydrate athletic products are high in HFCS and are therefore more likely to produce visceral obesity in athletes with insulin resistance.

## 16. Conclusions

Perhaps the important message from this narrative review is that there is little if any evidence that high-carbohydrate diets are essential for superior athletic performance [165,166]. I present the argument that all forms of exercise are ‘regulated’ and not ‘limited’ and that the key regulated variable especially during prolonged exercise appears to be the blood glucose concentration. The goal of such regulation is to forestall the possibility of hypoglycaemic brain damage.

Interestingly, mice that overexpress the protein targeting to glycogen specifically in the liver (PTG^OE^ mice) have superior endurance capacity despite unchanged skeletal muscle metabolism including unchanged muscle glycogen concentrations before and during exercise [180]. PTG^OE^ mice have increased liver glycogen stores and maintain higher blood glucose concentrations during prolonged exercise including in response to fasting.

The authors conclude that: ‘These results identify hepatic glycogen as a key regulator of endurance capacity in mice, an effect that may be exerted through the maintenance of blood glucose levels’ so that ‘In conclusion, these results identify hepatic glycogen as the key regulator of endurance performance in mice (current author’s added emphasis), an effect that may be exerted by the maintenance of blood glucose. Thus, in endurance sports such as marathon running and long-distance cycling, increased liver glycogen stores could maintain blood glucose and delay the onset of “hypoglycemia” or “hitting the wall” [180].

Importantly, in humans, quite small amounts of carbohydrate, ingested before or during exercise, are required to ensure that hypoglycemia does not develop during exercise. Ingesting more than that amount will not produce a superior outcome (Figure 29).

From the results of an extremely well-controlled study, Prins et al. [159] made the prediction that at least 88% of all the runners in the United States will not improve their 5 km running performances by following a high-carbohydrate diet. Thus, it is probable that the vast majority of recreational runners around the world will receive no benefit from eating the high-carbohydrate diets that are routinely prescribed for all [170,171,172].

Yet within that 88% of runners who will receive no benefit from the chronic ingestion of high-carbohydrate diets, there will be many who are insulin resistant and at risk of developing NAFLD and T2DM should they habitually eat that high-carbohydrate diet and include the regular consumption of HFCS-containing ‘sports’ drinks.

It is now high time to move away from the universal prescription of high-carbohydrate diets for all athletes regardless of their levels of athletic performance or insulin resistance on the false grounds that only high-carbohydrate diets will maximize athletic performance in all.

There is simply no good evidence to support the continued promotion of this advice.

Further, for those with insulin resistance, there is every reason to presume that following this advice will cause significant, avoidable, long-term harm.

## Figures and Tables

**Figure 1 nutrients-14-00862-f001:**
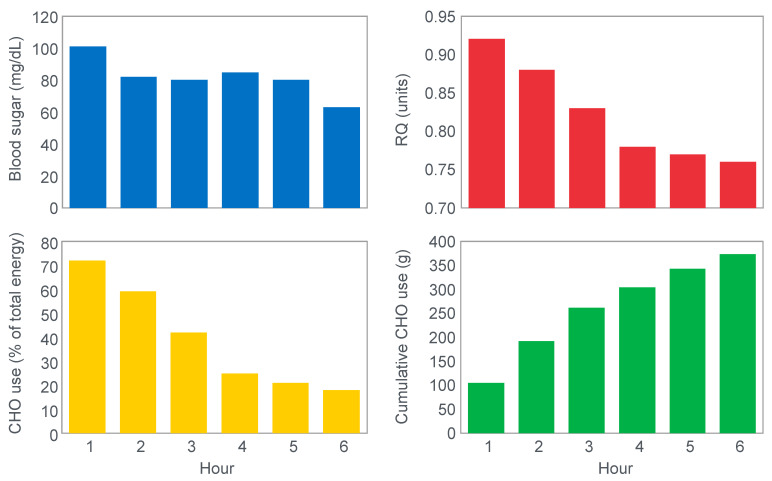
Changes in metabolic variables during 6 h of exercise in Runner Y. Reproduced from data in [12].

**Figure 2 nutrients-14-00862-f002:**
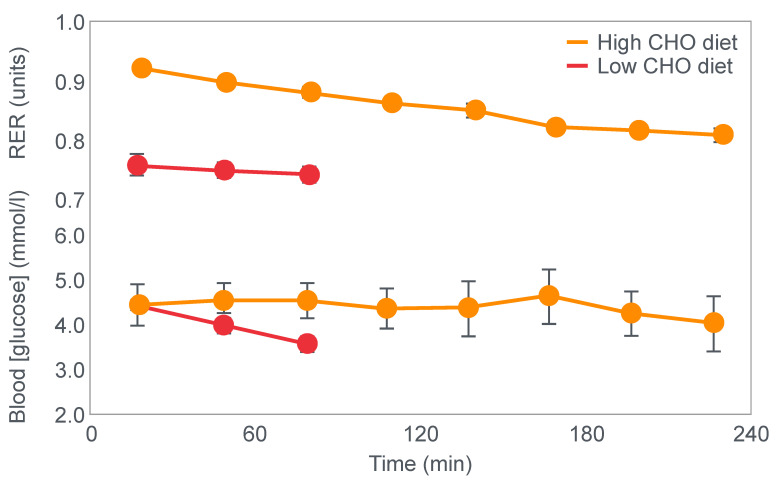
Changes in respiratory exchange ratio (RER) and blood glucose concentrations during prolonged exercise following high- and low-carbohydrate diets. Adapted with permission from ref. [14]. Copy-right 1939 Acta Physiologica (John Wiley and Sons).

**Figure 3 nutrients-14-00862-f003:**
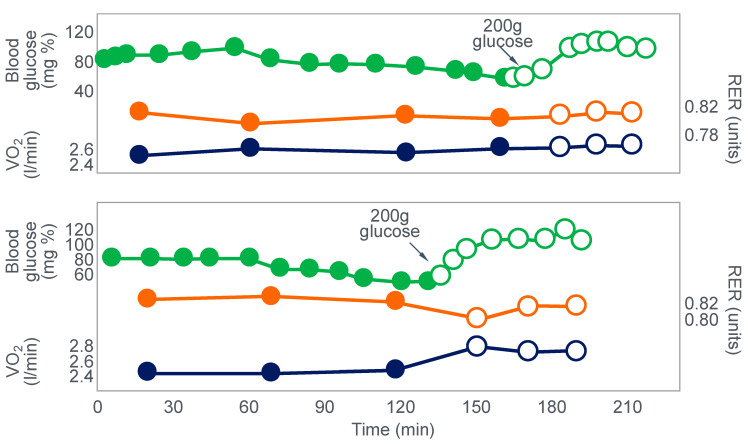
Effect on blood glucose concentrations, oxygen consumption (VO_2_) and respiratory exchange ratio (RER) of ingestion of 200 g of glucose at point of exhaustion in two athletes. Adapted with permission from ref. [14]. Copy-right 1939 Acta Physiologica (John Wiley and Sons).

**Figure 4 nutrients-14-00862-f004:**
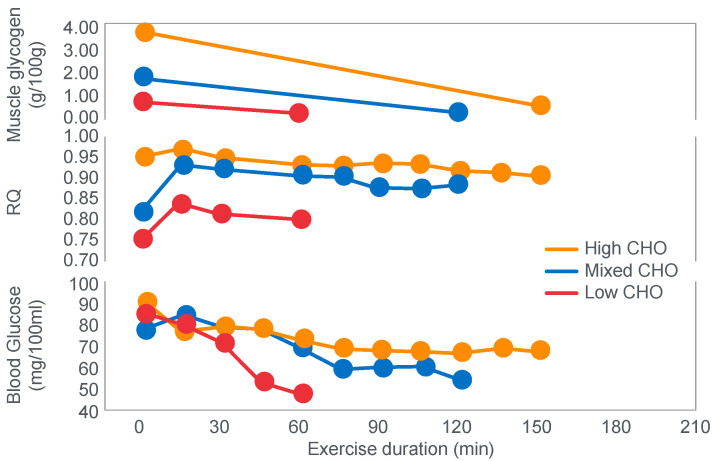
Changes in muscle glycogen concentrations, Respiratory Quotient (RQ) and blood glucose concentrations in subjects after acute adaptation to either high, mixed or low-carbohydrate diets. Adapted with permission from ref. [17]. Copy-right 1967 Acta Physiologica (John Wiley and Sons).

**Figure 5 nutrients-14-00862-f005:**
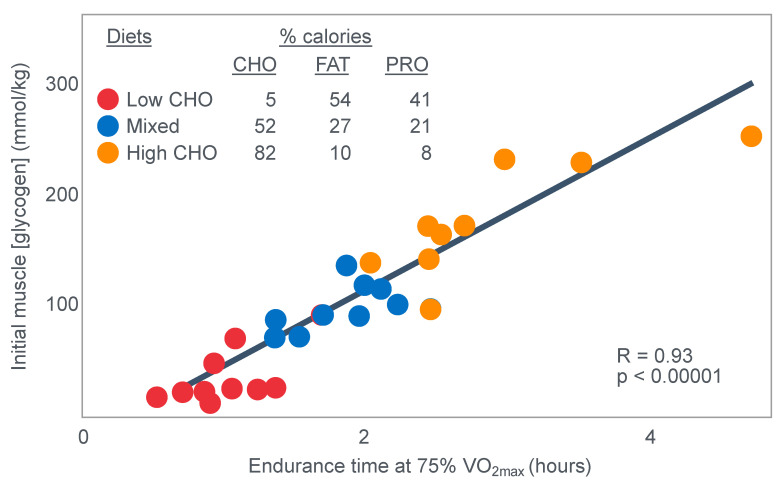
Relationship between initial muscle glycogen concentrations and exercise duration at 75%VO2max in subjects acutely adapted to high-, mixed- and low-carbohydrate diets. Adapted with permission from ref. [17]. Copy-right 1967 Acta Physiologica (John Wiley and Sons).

**Figure 6 nutrients-14-00862-f006:**
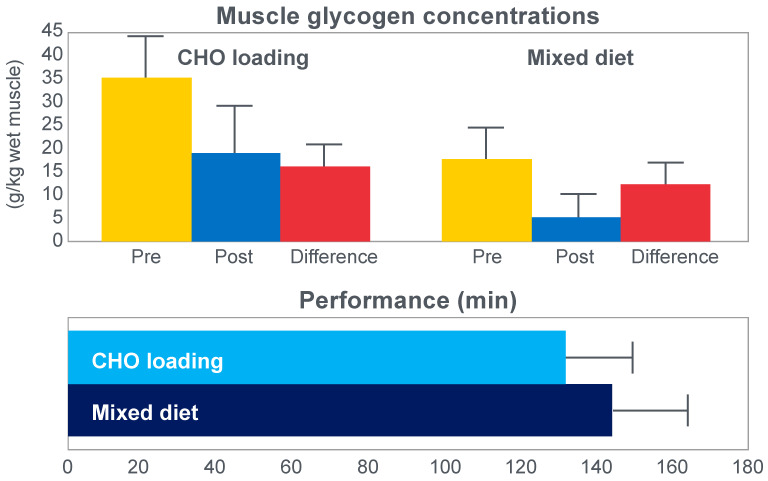
Changes in muscle glycogen concentrations and in running performances in athletes completing two 30 km races after high- (carbohydrate (CHO) loading) and mixed- carbohydrate diets. Drawn from data in [19].

**Figure 7 nutrients-14-00862-f007:**
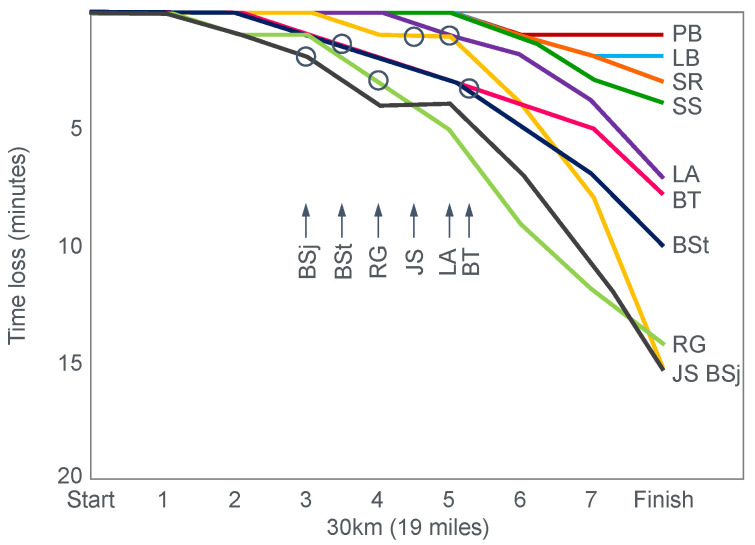
The time differences at 8 checkpoints when the same athletes ran two 30 km races after high- and mixed-carbohydrate diets. All athletes ran slower after the mixed-carbohydrate diet. Arrows indicate the point in the race at which athletes who started the race with low muscle glycogen concentrations, because they had eaten the mixed-carbohydrate diet, were predicted to have reached ‘limiting’ low muscle glycogen concentrations. Adapted with permission from ref. [19]. Copy-right 1971 American Physiological Society.

**Figure 8 nutrients-14-00862-f008:**
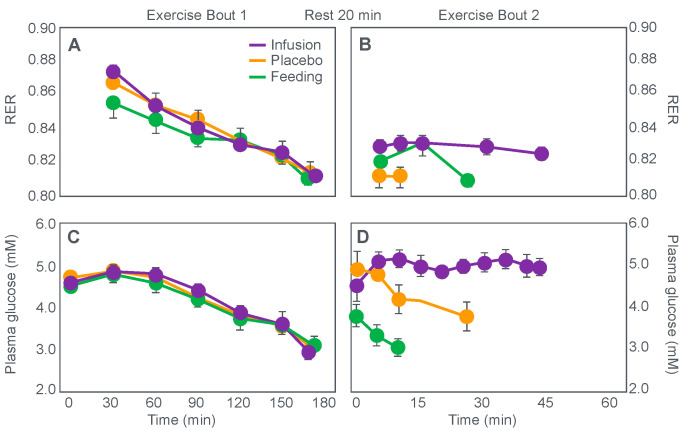
Changes in respiratory exchange ratio (RER) (panel **A**) and in plasma glucose concentrations (panel **C**) in athletes during an initial bout of 180 min of exercise (Exercise Bout 1) that induced profound hypoglycemia in all the athletes. (Panels **B**,**D**) show changes in these variables in three groups of athletes who received either a placebo or carbohydrate by mouth, or a continuous glucose infusion during Exercise Bout 2 that followed 20 min after the conclusion of Exercise Bout 1. Adapted with permission from ref. [27]. Copy-right 1987 American Physiological Society.

**Figure 9 nutrients-14-00862-f009:**
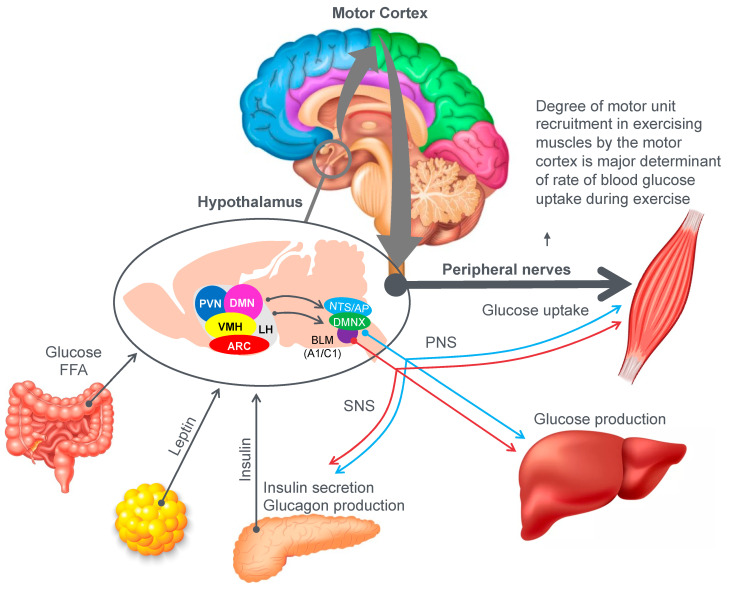
From [46]. Specialized brain areas in the hypothalamus and brain stem (AP, area postrema; ARC, arcuate nucleus; BLM, basolateral medulla; DMN, dorsomedial nucleus; DMNX, dorsal motor nucleus of the vagus; LH, lateral hypothalamus; NTS, nucleus of the solitary tract; PNS, parasympathetic nervous system; PVN, paraventricular nucleus; SNS, sympathetic nervous system; VMH, ventromedial hypothalamus) sense peripheral metabolic signals through hormones and nutrients to regulate whole body glucose metabolism. The autonomic nervous system contributes by modulating pancreatic insulin/glucagon secretion, hepatic glucose production and skeletal muscle glucose uptake. During exercise, the major threat to blood glucose homeostasis is the rate of blood glucose uptake by the exercising muscles. It therefore makes sense that the hypothalamic regulators of blood glucose homeostasis must also influence the degree of motor unit recruitment that is allowed in the exercising limbs, specifically to ensure that hypoglycaemic brain damage does not occur during especially prolonged exercise. Adapted, modified and redrawn from the original in [46] with the addition of the action of the hypothalamic → motor cortex → spinal cord → peripheral nerve → skeletal muscle homeostatic reflex control.

**Figure 11 nutrients-14-00862-f011:**
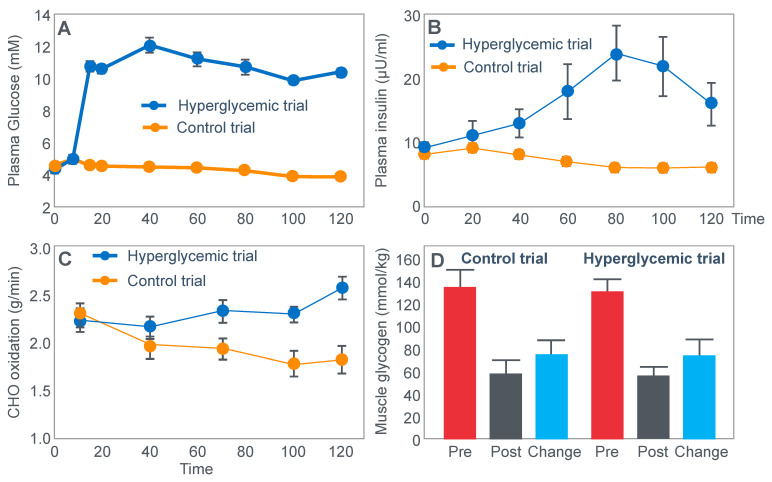
Changes in plasma glucose (panel **A**) and plasma insulin concentrations (panel **B**), in rates of carbohydrate oxidation (panel **C**) and in muscle glycogen concentration before and after exercise, including muscle glycogen change (panel **D**), in athletes who exercised for 120 min with or without a glucose infusion that produced profound hyperglycemia. Note that unlike the response shown in Figure 10, subjects in this study did not develop hypoglycemia during the control condition even though they had fasted for 12–14 h before exercise but had eaten a high-carbohydrate diet. Adapted with permission from ref. [64]. Copy-right 1991 American Physiological Society.

**Figure 12 nutrients-14-00862-f012:**
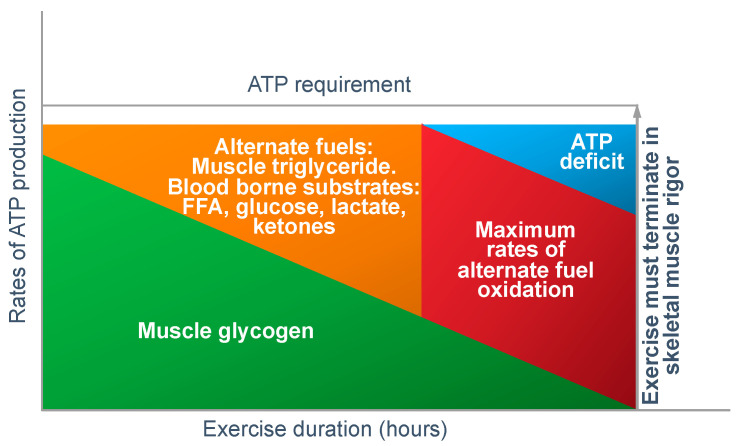
If an inability to generate sufficient ATP in the exercising muscles was the key factor ‘limiting’ exercise performance during prolonged exercise then, in this unregulated system, the sole outcome would always be the development of skeletal muscle rigor. Since skeletal muscle rigor does not happen during any form of exercise, skeletal muscle function (and metabolism) must be homeostatically regulated, specifically to prevent this catastrophic outcome.

**Figure 13 nutrients-14-00862-f013:**
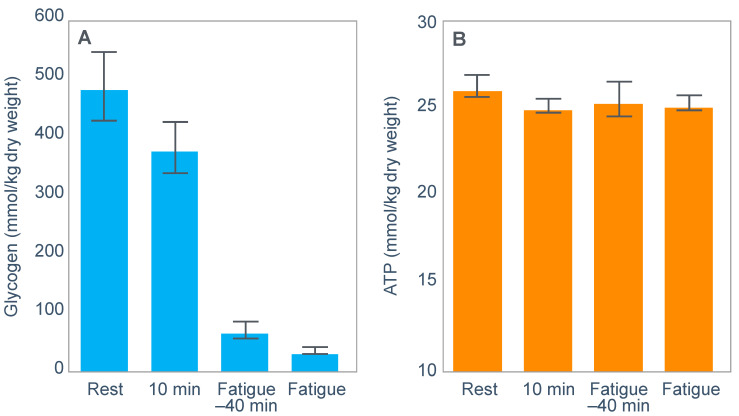
Muscle glycogen concentrations (panel **A**) were reduced during exercise, reaching very low levels at fatigue. Yet muscle ATP concentrations (panel **B**) remained at pre-exercise resting concentrations during exercise and at the point of fatigue. The total adenine nucleotide pool (TAN) was also not reduced at fatigue. Adapted with permission from ref. [99]. Copy-right Year 2003 American Physiological Society.

**Figure 14 nutrients-14-00862-f014:**
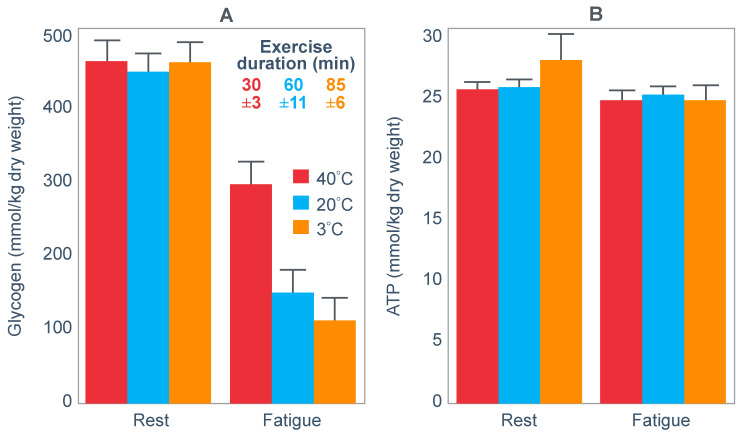
Although muscle glycogen concentrations (panel **A**) were reduced during exercise, they were not similar at the point of exercise termination in three different conditions of external heat (40, 30 and 3 °C). At the point of fatigue, muscle ATP concentrations (panel **B**) were not different from pre-exercise resting concentrations in all three conditions. Adapted with permission from ref. [101]. Copy-right 1999 Acta Physiologica Scandinavica.

**Figure 15 nutrients-14-00862-f015:**
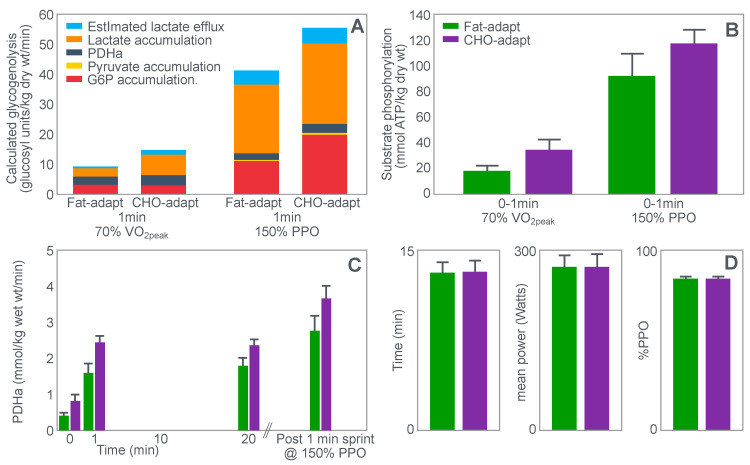
Changes in calculated glycogenolytic rates (panel **A**), substrate phosphorylation (panel **B**), and PDHa concentration (panel **C**) during exercise at 70%VO2max and after a 1 min sprint at 150% peak power output (PPO) in subjects following short-term adaptation to high-carbohydrate or high-fat diets. Exercise performance during the self-paced time trial at ~90%VO2max—time to complete the time trial; mean power and % PPO was not influenced by the diet (panel **D**). Adapted with permission from ref. [43]. Copy-right 2006 American Physiological Society.

**Figure 16 nutrients-14-00862-f016:**
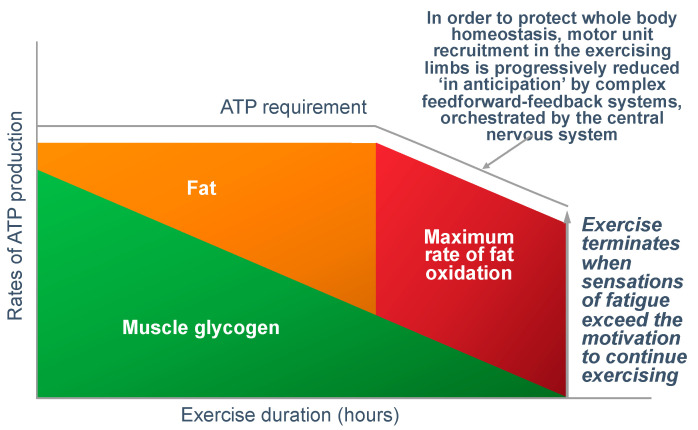
The extent of motor unit recruitment in the muscles in the exercising limbs determines the athlete’s exercise intensity. This also determines the metabolic response within those muscles. Thus, the most appropriate method to protect the muscles from a catastrophic metabolic outcome (Figure 12) is to regulate the number of motor units that are allowed to be activated in the muscles in the exercising limbs. This model therefore predicts that if the ability to generate ATP in the exercising limbs is restrained (‘limited’) for any reason, the immediate response of the central nervous system will be to reduce the number of motor units it chooses to recruit in those active muscles. This will ensure that homeostasis is maintained. Note this diagram suggests that exercise terminates only after muscle glycogen is completely depleted. In fact, the central governor ensures that exercise always terminates before this point is reached.

**Figure 17 nutrients-14-00862-f017:**
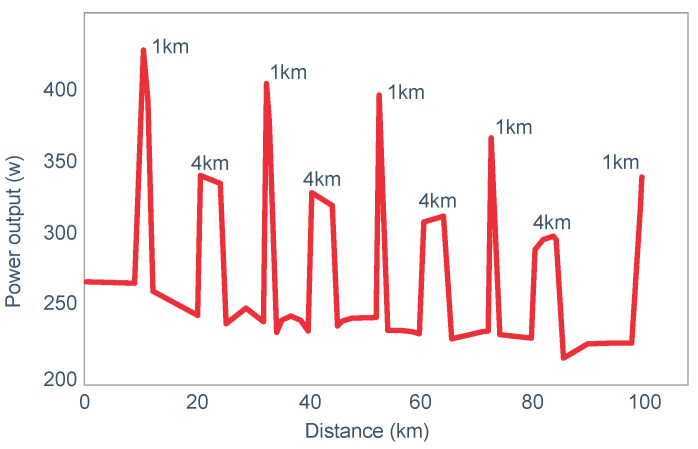
Power outputs measured in a series of 1 and 4 km sprints during a 100 km cycling time trial in the laboratory, fell progressively. This reduction could be due either to mechanisms present in the exercising muscles (peripheral fatigue) or the result of reduced motor unit recruitment in the exercising muscles by the motor cortex in the brain (central regulation).

**Figure 18 nutrients-14-00862-f018:**
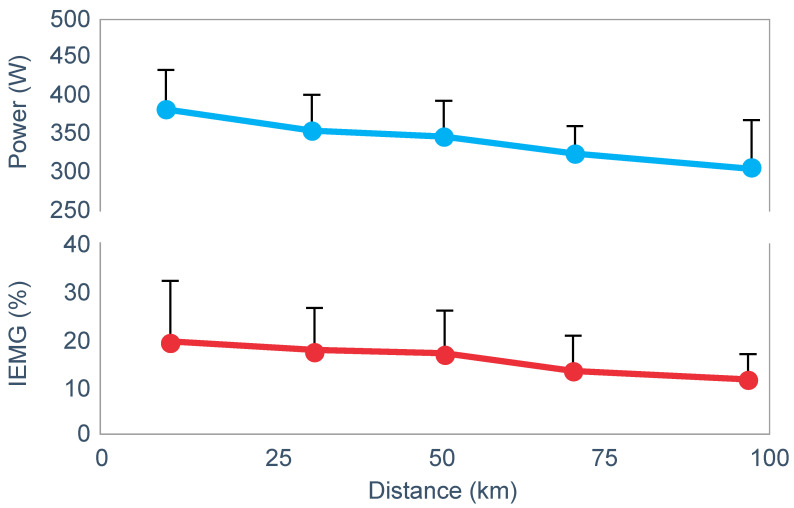
Equivalent and progressive reductions in power output and electromyographic activity in the exercising leg muscles during five 1 km sprint during a 100 km cycling time trial in the laboratory. This indicates the presence of a centrally-regulated homeostatic control that progressively reduces motor unit recruitment in the exercising limbs during prolonged exercise. This shows that performance is ‘regulated’, not ‘limited’ during exercise [11]. Adapted with permission from ref. [110]. Copy-right 2001 American Physiological Society.

**Figure 19 nutrients-14-00862-f019:**
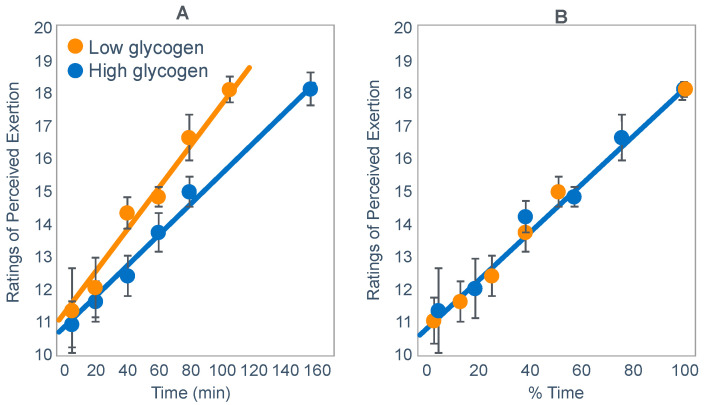
Ratings of perceived exertion (RPE) rose as a linear function of absolute exercise durations in the study of Baldwin et al. [42] in subjects who began exercise with either low or high muscle glycogen content (panel **A**). When the same data are expressed as a function of relative (%) time, (panel **B**) the lines overlap. Thus, the RPE rises as a linear function of the expected duration of the exercise that is being performed. Adapted with permission from ref. [116]. Copy-right Year 2004 American Physiological Society.

**Figure 20 nutrients-14-00862-f020:**
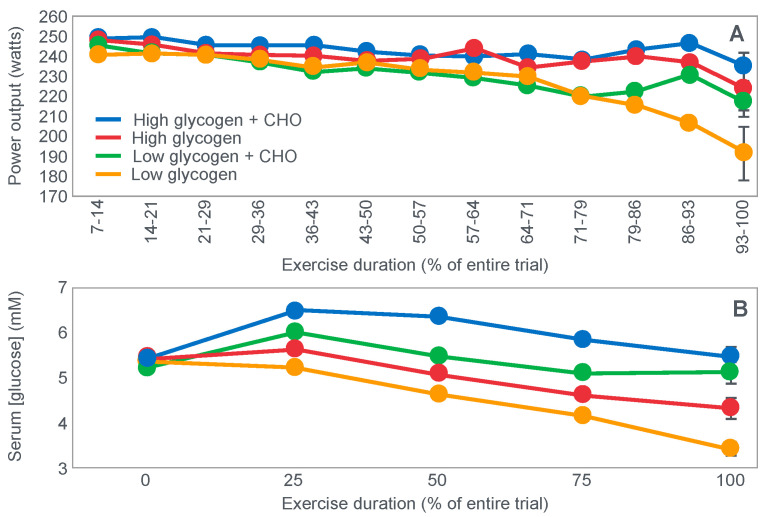
Power outputs (panel **A**) and serum glucose concentrations (panel **B**) in subjects performing a 70 km cycling trial under four different conditions of starting muscle glycogen contents (high or low) and ingestion during exercise (placebo or 9% carbohydrate solution). Adapted with permission from ref. [71]. Copy-right Year 1993 American Physiological Society.

**Figure 21 nutrients-14-00862-f021:**
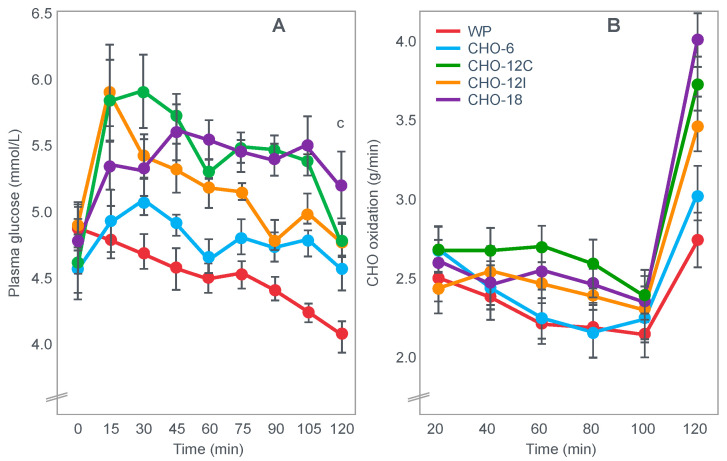
Changes in plasma glucose concentrations (panel **A**) and in rates of carbohydrate oxidation (panel **B**) when subjects drank either placebo or one of four different carbohydrate solutions with different carbohydrate concentrations (6, 12 or 18%). CHO-12C and CHO-12I refer to trials that were conducted continuously (C) or intermittently (I). Adapted with permission from ref. [51]. Copy-right 1989 American Physiological Society.

**Figure 22 nutrients-14-00862-f022:**
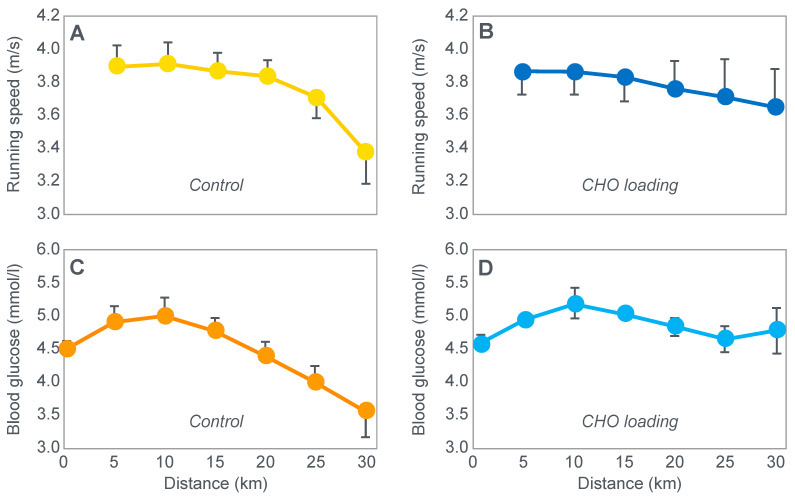
Changes in running speeds (panels **A**,**B**) and in blood glucose concentrations (panels **C**,**D**) in subjects running a 30 km time trial before (panels **A**,**C**) and after (panels **B**,**D**) 7 days of ‘carbohydrate loading’. ‘Carbohydrate loading’ prevented the development of hypoglycemia during the final 10 km of the race (panel **D**) and was associated with better maintenance of running speed (panel **B**). Adapted with permission from ref. [72]. Copy-right 1992 Springer Nature (European Journal of Applied Physiology).

**Figure 23 nutrients-14-00862-f023:**
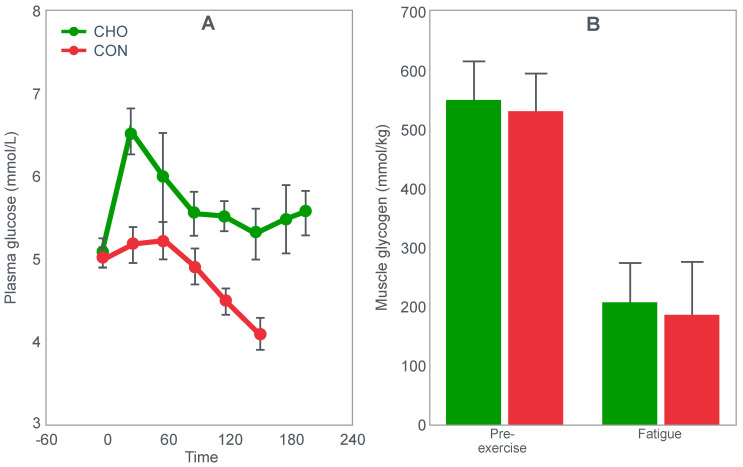
Changes in plasma glucose (panel **A**) and muscle glycogen concentrations (panel **B**) in subjects performing prolonged exercise with either carbohydrate (CHO) or placebo (CON) ingestion during exercise. Adapted with permission from ref. [57]. Copy-right 1999 American Physiological Society.

**Figure 24 nutrients-14-00862-f024:**
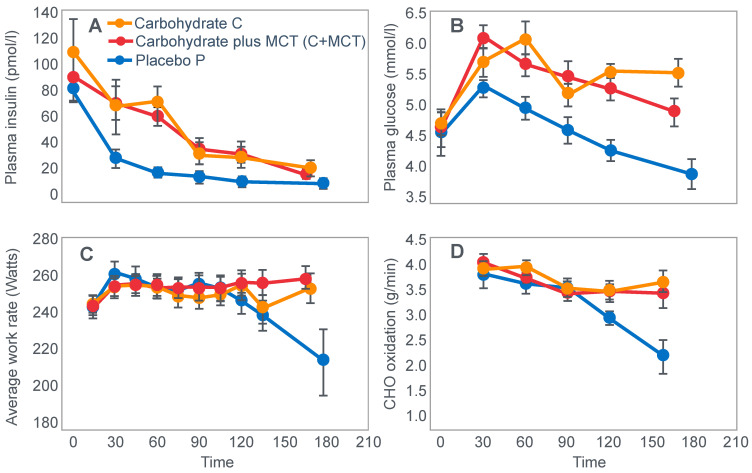
Changes in plasma insulin (panel **A**) and plasma glucose (panel **B**) concentrations; average work rate (panel **C**) and rates of carbohydrate oxidation (panel **D**) when subjects ingested either carbohydrate (C), carbohydrate plus medium chain triglycerides (C + MCT) or a sweetened placebo (P) during a 100 km cycling time trial in the laboratory. Note: In the original diagrams, it is not possible in some panels to distinguish between lines drawn for the C and C + M groups as the lines appear to be incorrectly labelled. However, the lines are largely overlapping. Adapted with permission from ref. [69]. Copy-right 2000 American Physiological Society.

**Figure 25 nutrients-14-00862-f025:**
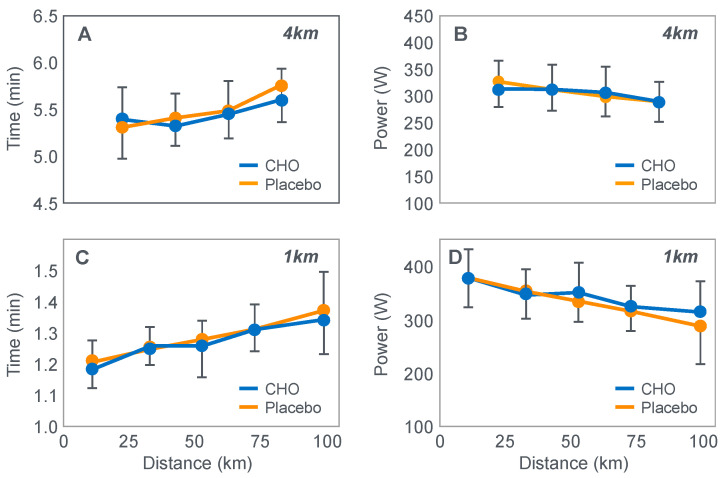
Times (panels **A**,**C**) and power outputs (panels **B**,**D**) in 4 km (panels **A**,**B**) and 1 km sprints (panels **C**,**D**) interspersed within a 100 km cycling time trial were not different when subjects had undergone pre-exercise ‘carbohydrate loading’ with either real carbohydrates or a placebo. Adapted with permission from ref. [120]. Copy-right 2000. American Physiological Society.

**Figure 26 nutrients-14-00862-f026:**
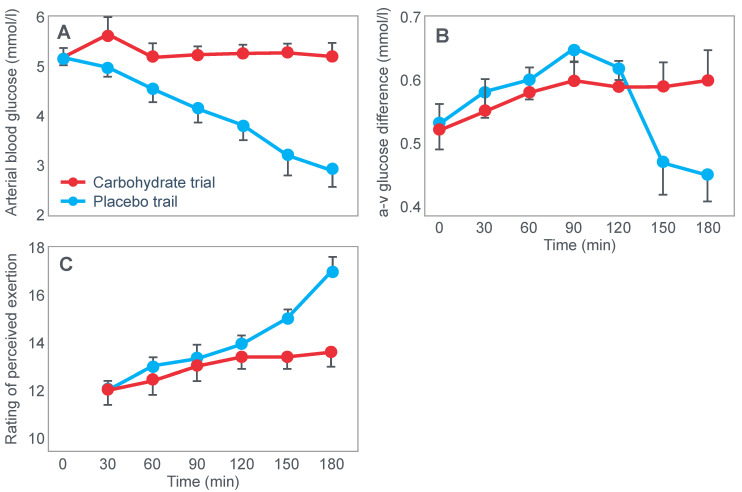
Changes in arterial blood glucose (panel **A**) concentrations, arteriovenous glucose differences across the brain (panel **B**) and ratings of perceived exertion (panel **C**) in subjects when they ingested placebo or glucose during 180 min of exercise. Adapted with permission from ref. [123]. Copy-right 2003 Acta Physiologica (John Wiley and Sons).

**Figure 27 nutrients-14-00862-f027:**
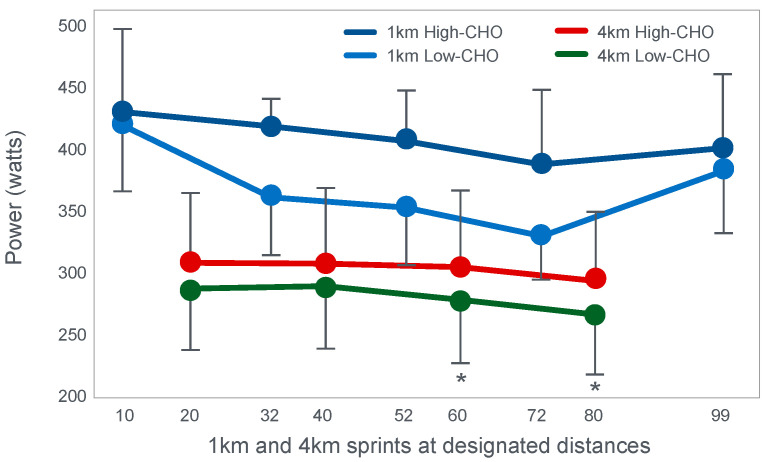
Power outputs during either 1 or 4 km sprints interspersed within a 100 km laboratory cycling time trial when athletes ate either the low-carbohydrate or high-carbohydrate diets for six days followed by a single day of ‘carbohydrate loading’. Adapted with permission from ref. [104]. Copy-right 2006 American Physiological Society.

**Figure 28 nutrients-14-00862-f028:**
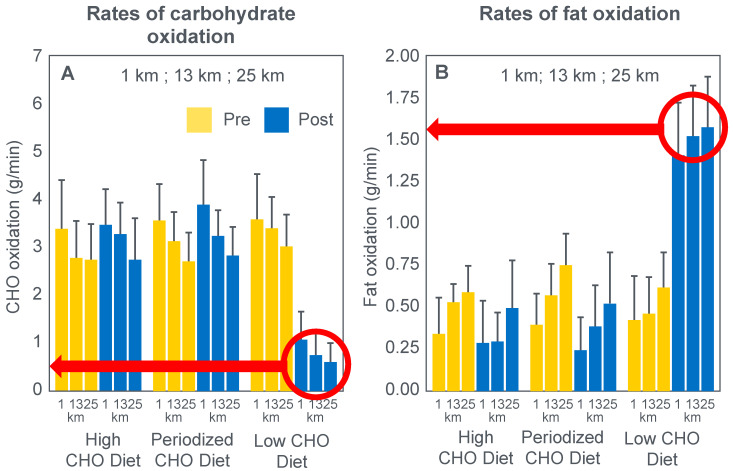
Rates of carbohydrate (panel **A**) and fat oxidation (panel **B**) measured at 1, 13 and 25 km in race walkers during two 25 km time trials before and after they had adapted to one of three different dietary options—high-carbohydrate diet; a periodized carbohydrate diet; and a low-carbohydrate high-fat diet. Adapted with permission from ref. [134]. Copy-right 2017 The Physiological Society (John Wiley and Sons).

**Figure 29 nutrients-14-00862-f029:**
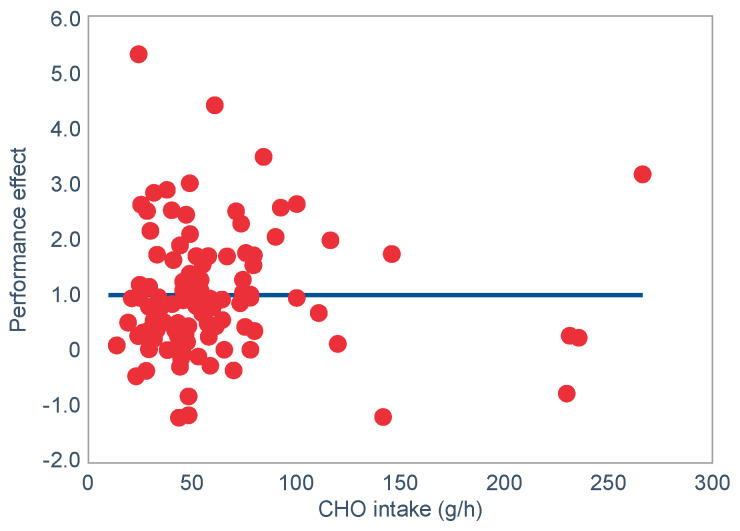
The absence of any relationship between the amount of carbohydrate ingested per hour during exercise and any measured performance benefit. Adapted with permission from ref. [139]. Copy-right 2010 Springer Nature.

## Data Availability

All the data used in this review are from the original published studies and are available within the cited references.

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
