# Peer review of "What Is the Evidence That Dietary Macronutrient Composition Influences Exercise Performance? A Narrative Review"

_nutrients, 2022, doi:10.3390/nu14040862_

Round 1
Reviewer 1 Report
What is the evidence that dietary macronutrient composition influences exercise performance? A Narrative review.
This is an interesting and extensive narrative review that reflects on the role of carbohydrate/muscle glycogen oxidation in endurance performance.
The author states that ‘brainless’ models cannot explain why fatigue rather than skeletal muscle rigor is the outcome of prolonged exercise. It has been reasoned that when muscle glycogen stores are emptied, the CHO oxidation is no longer able to support ATP generation so that ATP levels should drop and lead to skeletal muscle rigor after prolonged exercise. The absence of dropping ATP levels with prolonged exercise (Figures 13 and 14) are used as evidence for rejecting these ‘brainless’ models. However, these figures also demonstrate that glycogen storage is not zero, implying the sustained ATP levels could still be accommodated by oxidation of the (remaining) glycogen. Please elaborate.
It is clear that muscle glycogen or carbohydrate oxidations plays an essential role in exercise of prolonged duration. However, some claims seem to be a bit overstated, such as “the single finding that transformed the exercise sciences was that muscle glycogen content…” or “other than the athlete’s natural ability and degree of endurance training, the athlete’s pre-exercise muscle glycogen content is the single most important determinant...”. Of course, this aspect played in important part in the exercise sciences, but the field is much larger than muscle glycogen alone. Please add more nuance to the manuscript.
Of note, the formatting of the manuscript could be improved. Some of the figures are a bit unclear, with multiple axes that are not clearly separated. In addition, figure legends or captions appear within the text of the manuscript, which does hamper readability of the manuscript to some extent.
Author Response
Thank you for your comments which are appreciated.
Your first point about the diagrams showing complete muscle glycogen depletion is correct. Complete muscle glycogen depletion does not occur – there is always some glycogen left at exhaustion as shown in other of the figures. Yet continued exercise does not further lower those muscle glycogen stores as described in the text. Also fatigue occurs at higher muscle glycogen levels in those eating high carbohydrate diets than in those on low carbohydrate diets; again suggesting that something else is stopping the exercise, rather than an absence of muscle glycogen. I’ve added a few lines of text to accommodate your astute observation.
Your second point is that some of the statement regarding muscle glycogen are overstated and that the article should be more nuanced. I disagree. This article is written for a journal interested in nutrition not in other determinants of exercise performance. When one reads the sports nutrition literature it is clear that athletes still believe and are taught that carbohydrates are crucial for performance because they increase muscle glycogen content etc.
I stand by the other statements as I was around in 1967 when these ideas were first promoted and especially the sports nutrition industry has continued to drive this message ever since. It seems time that they need to be properly refuted.
Many thanks for your comments.
Reviewer 2 Report
This review article doesn't follow the scientific base of evidence. The author uses evidence to support his hypothesis, and fails to acknowledge the entire results and/or consensus amongst scientists. Turning controversy into arguments that go against consensus isn't good science. The author goes to many lengths to back up his theories, but fails to make any sense with his arguments.
Author Response
The reviewer argues that it is not a good idea to go against the scientific consensus as this is not “good science”. The reviewer states that I do not make sense with my arguments but does not explain where the article fails to make sense.
I argue that if one believes the consensus to be wrong then it is one’s duty to point out where the consensus may be wrong and to provide an alternative.
Which is what I have done.
Reviewer 3 Report
Review of manuscript -Manuscript ID: nutrients-1507035
The paper presents an interesting review of research on the influence of diets with different macronutrient content on exercise results. The authors carry out the analysis depending on the possibility of conducting research in a given historical period. They interpret research from laboratory studies to develop a needle-needle muscle biopsy technique to measure muscle glycogen treatment, to studies on the potentially negative long-term health effects of high-carbohydrate diets in people with insulin resistance.
The title encourages you to read the content of the article.
The summary is complete and meets the requirements.
The purpose of the publication was clearly defined.
The presentation methods are appropriate.
The authors refer to a large number of items in the bibliography (180). This is actually a lot and little, for a trial, such an extensive topic (90 years of research). However, the authors emphasize in their critical study that some studies are missing, there are few, or the conclusions are drawn depending on the adopted model/concept of the study.
The results of the survey rightly emphasize how subsequent sureger results were influencing not necessarily the right solutions used in practice.
A critical review of the presented research confirms the assumptions of the work as well as summarizes what seems to be the most important and safe to use, according to the authors.
The work is interesting, important.
Author Response
Thank you for your helpful comments. I appreciate it that you feel the article is of value. I have not made any changes based on your comments.
Round 2
Reviewer 2 Report
The manuscript, as written, convey misinformation leading the reader to become confused following the authors thoughts down a rabbit hole.
Author Response
The reviewer's comment is: "The manuscript, as written, convey misinformation leading the reader to become confused following the authors thoughts down a rabbit hole".
However the reviewer does not specify what misinformation he refers to. All the points I made in the review are fully supported by the cited literature.
I argue that the source of the current "misinformation" are the ideas which I show to be not scientifically supported.
Suggesting that the article leads the reader down a "rabbit hole" is only valid if the reviewer can provide evidence that I have done this.
I am very happy with the logic that I have presented. It reflects more than 40 years thinking about and researching the topic. In fact only when I had written the article did I begin to realise the extent of the contribution our research team has made to this topic.
That expertise on this topic is not widely available in the sports sciences.